# PHF6 promotes non-homologous end joining and G2 checkpoint recovery

Daniël O Warmerdam[1,2,*,†] , Ignacio Alonso-de Vega[3,4,†], Wouter W Wiegant[5], Bram van den Broek[2,6], Magdalena B Rother[5], Rob MF Wolthuis[7], Raimundo Freire[3,4,8], Haico van Attikum[5] , René H Medema[2] & Veronique AJ Smits[3,4,8,**]

## Abstract

The cellular response to DNA breaks is influenced by chromatin compaction. To identify chromatin regulators involved in the DNA damage response, we screened for genes that affect recovery following DNA damage using an RNAi library of chromatin regulators. We identified genes involved in chromatin remodeling, sister chromatid cohesion, and histone acetylation not previously associated with checkpoint recovery. Among these is the PHD finger protein 6 (PHF6), a gene mutated in Börjeson–Forssman–Lehmann syndrome and leukemic cancers. We find that loss of PHF6 dramatically compromises checkpoint recovery in G2 phase cells. Moreover, PHF6 is rapidly recruited to sites of DNA lesions in a PARP-dependent manner and required for efficient DNA repair through classical non-homologous end joining. These results indicate that PHF6 is a novel DNA damage response regulator that promotes end joining-mediated repair, thereby stimulating timely recovery from the G2 checkpoint.

**Keywords** checkpoint recovery; chromatin regulators; NHEJ; PHF6; siRNA screen
**Subject Category** DNA Replication, Recombination & Repair

## Introduction

Genome instability is a hallmark of cancer cells and a driving force in tumor progression [1]. Various forms of DNA damage can lead to genomic alterations, but double-strand breaks (DSBs) are particularly deleterious as they can lead to a significant loss of genetic material or result in chromosomal translocations [2]. Mammalian cells have developed a complex response to DNA damage that facilitates the repair of damaged DNA and prevents genomic alterations [3]. This response is directed by activation of the DNA damage response (DDR) kinases ATM and ATR, which phosphorylate numerous downstream targets such as the checkpoint kinases Chk2 and Chk1, resulting in a transient cell cycle arrest that allows time for DNA repair [4,5].

Double-strand breaks are repaired through direct ligation of the broken DNA ends in a process called non-homologous end joining (NHEJ) or through homologous recombination (HR), which uses the sister chromatid as a repair template [6]. DSB repair is differentially regulated throughout the cell cycle. While NHEJ is the dominant pathway in G1 phase, HR is predominantly active in S/G2 phase when sister chromatids are available [7]. DDR proteins 53BP1 and BRCA1 are recruited to the sites of DNA lesions and play a critical role in DSB repair pathway choice. Whereas 53BP1 negatively regulates DNA end resection, a key step in the initiation of HR, and thus promotes repair through NHEJ, BRCA1 promotes DNA end resection and thereby the removal of 53BP1 from sites of DNA damage and in that way directs repair of DSBs toward HR [8].

Once the DNA damage is repaired, cell cycle progression is resumed in a process known as checkpoint recovery [9]. Resuming cell cycle progression after DNA damage in G2 phase is an active process dependent on several kinases, including PLK1 and its upstream regulators Aurora A and Bora, that re-activate cyclin B1/Cdk1 [10]. Together with the βTrCP-SCF ubiquitin ligase, PLK1 also directs the degradation of checkpoint mediator Claspin and thereby mediates Chk1 silencing [11–13]. Additionally, the phosphatase PPM1D contributes to checkpoint recovery by counteracting DNA damage-induced phosphorylation events, including the phosphorylation of important players in the DDR activation, such as histone H2AX, p53, and Chk1 [14–16].

1  CRISPR Platform, Cancer Center Amsterdam, Amsterdam UMC, University of Amsterdam, Amsterdam, The Netherlands
2  Division of Cell Biology, Oncode Institute, The Netherlands Cancer Institute, Amsterdam, The Netherlands
3  Unidad de Investigación, Hospital Universitario de Canarias, La Laguna, Tenerife, Spain
4  Instituto de Tecnologías Biomédicas, Universidad de La Laguna, Tenerife, Spain
5  Department of Human Genetics, Leiden University Medical Center, Leiden, The Netherlands
6  BioImaging Facility, The Netherlands Cancer Institute, Amsterdam, The Netherlands
7  Section of Oncogenetics, Department of Clinical Genetics, Vrije Universiteit Amsterdam, Cancer Center Amsterdam, Amsterdam UMC, Amsterdam, The Netherlands
8  Universidad Fernando Pessoa Canarias, Las Palmas de Gran Canaria, Spain
   *Corresponding author. Tel: +34 9226 78107; E-mail: d.o.warmerdam@amsterdamumc.nl
   **Corresponding author. Tel: +31 2056 64671; E-mail: vsmits@ull.edu.es
   †These authors contributed equally to this work

Over the last decades, we gained a detailed understanding of the mechanisms that detect and repair DNA damage and those that inhibit the cell cycle machinery leading to checkpoint arrest [6]. However, it remains largely unclear how the DDR, and particularly the completion of DNA repair, is coupled to checkpoint recovery.

It is evident that chromatin plays an important role in regulating the DDR [14,17–19]. In response to DNA damage, histones undergo different post-translational modifications, including phosphorylation, acetylation, methylation, and ubiquitination. Some of these modifications, such as histone acetylation, also locally relax the compacted chromatin structure and physically facilitate the accessibility of repair proteins to the lesion [20]. In addition, ATP-dependent remodelers modify damaged chromatin by changing nucleosome position or composition. An increasing number of chromatin modifications and remodelers have been implicated in the regulation of checkpoint activation and DNA repair. Many proteins involved in the DDR use the damaged and/or modified chromatin as a template for their actions, and chromatin modifications have been described to specifically recruit repair factors [21–23]. Interestingly, defects in the regulation of chromatin modulators are associated with genomic instability and cancer development [24]. Although histone modifications are likewise thought to be important to restore the original chromatin structure after repair of the lesion and to resume cell cycle progression [19,22], there is little evidence for the direct implications of chromatin modifications in checkpoint recovery. Here, we applied a reverse genetic approach to discover novel chromatin regulators required for the DDR by measuring their impact on recovery from the G2 checkpoint and identified PHF6 as a new DNA repair and checkpoint recovery factor.

## Results and Discussion

To identify chromatin-associated regulators involved in the recovery from the G2 checkpoint, we performed an RNA interference screen in human U2OS cells. An siRNA-based library targeting 529 genes related to the structure, maintenance, or modification of chromatin [25] was used to screen for the efficiency of cell cycle committed cells to recover after treatment with a non-lethal dose of ionizing radiation (IR). Cells, synchronized in G2, were irradiated, and mitotic entry was determined using immunofluorescence analysis of phosphorylated histone H3 (Fig 1A). Results were examined using a Z-score approach [26]. A cut-off value of 2 was chosen for genes

which knockdown increased recovery and 1 for genes that resulted in decreased recovery after IR. Knockdown of PPM1D and βTrCP1/2, which were previously described to be involved in checkpoint recovery [11,16] and were added to the library as positive controls, significantly affected recovery, illustrating the feasibility and validity of our screen. Depletion of an additional 28 genes resulted in a reproducible reduction in recovery, whereas the knockdown of 13 genes caused increased recovery (Fig 1B and C). Upon rescreening of these 41 hits using the four individual siRNA oligonucleotides, 36 genes gave rise to altered recovery with at least 2 out of 4 siRNA oligos (Fig 1C). These genes were selected for further analysis. TP53BP1 was excluded for further analysis, as it has a known role in the DDR and it is therefore not surprising that knockdown of 53BP1 has an effect on checkpoint recovery [27]. After selecting the siRNA oligonucleotide sequence that most strongly influenced recovery in the primary screen, their impact on checkpoint recovery was determined using an independent FACS-based assay after IR treatment in both U2OS (Fig 1D) and non-transformed RPE1 cells (Fig 1E). We scored effects on mitotic entry measured by MPM2 positivity of the 36 genes identified in the original screen and found 22 genes contributed in a similar manner to checkpoint recovery in both U2OS and RPE1. However, many of the hits that led to increased recovery in U2OS showed an opposite response in RPE-1 cells. We subsequently focused on the 22 genes affecting mitotic entry in both cell lines (Table EV1).

Pathway analysis of these hits indicated a strong functional correlation between the identified genes (Fig 1F). Nine genes required for recovery are associated with the NuA4 protein complex. This ATPase-dependent chromatin-remodeling complex with additional histone acetylation activity regulates nucleosome stability and modulates chromatin structure. Five additional candidates are connected to the cohesin complex, which is required for sister chromatid cohesion. These results suggest a functional relationship between these two protein complexes and checkpoint recovery. In accordance with our data, NuA4-mediated chromatin-remodeling activities like nucleosome assembly/disassembly and specific histone deposition/modifications as well as sister chromatid cohesion have shown to be important for DNA damage checkpoint signaling, DNA repair and checkpoint recovery, thereby stressing the importance of these protein complexes in the DDR and additionally demonstrating the feasibility of our screening setup [28–34]. Moreover, eight other genes not described to be part of these or other shared complexes were identified as regulators of DNA damage recovery: SETDB2, PHF3, RBBP6, PHF6, MTA1, PHF12, MLL2, and LIN9.

**Figure 1.  RNA interference-based screen for G2 checkpoint recovery.**

A   Experimental setup used to screen for chromatin regulators involved in the IR-induced G2 checkpoint response and recovery. Unless stated otherwise, 5 Gy of IR was used.

B   Results of three independent microscopy-based screens in U2OS cells. Knockdown of luciferase (blue) and the positive controls PPM1D and βTrCP (red) are indicated. A low/high *Z*-score indicates less/more phospho-histone H3 positive cells compared with the control, reflecting reduced/increased numbers of cells going into mitosis in the respective knockdowns after 5 Gy of IR.

C   Deconvolution of the 41 primary hits in U2OS cells by four separate siRNA oligonucleotides. Gray represents a recovery value that is similar to the luciferase control (< 0.5 SD), and the red represents an altered recovery value (> 0.5 SD) in two independent experiments.

D, E   U2OS cells (D) and RPE1 hTert immortalized primary epithelial cells (E) were transfected with siRNA oligos targeting the hits in a similar setup as described in A), but now analyzed for mitotic entry by MPM2 and PI staining by flow cytometry. For normalization, see Materials and Methods. Error bars represent the SEM of three independent experiments.

F   Graphical explanation of the known interactions between the identified hits through STRING pathway analysis.

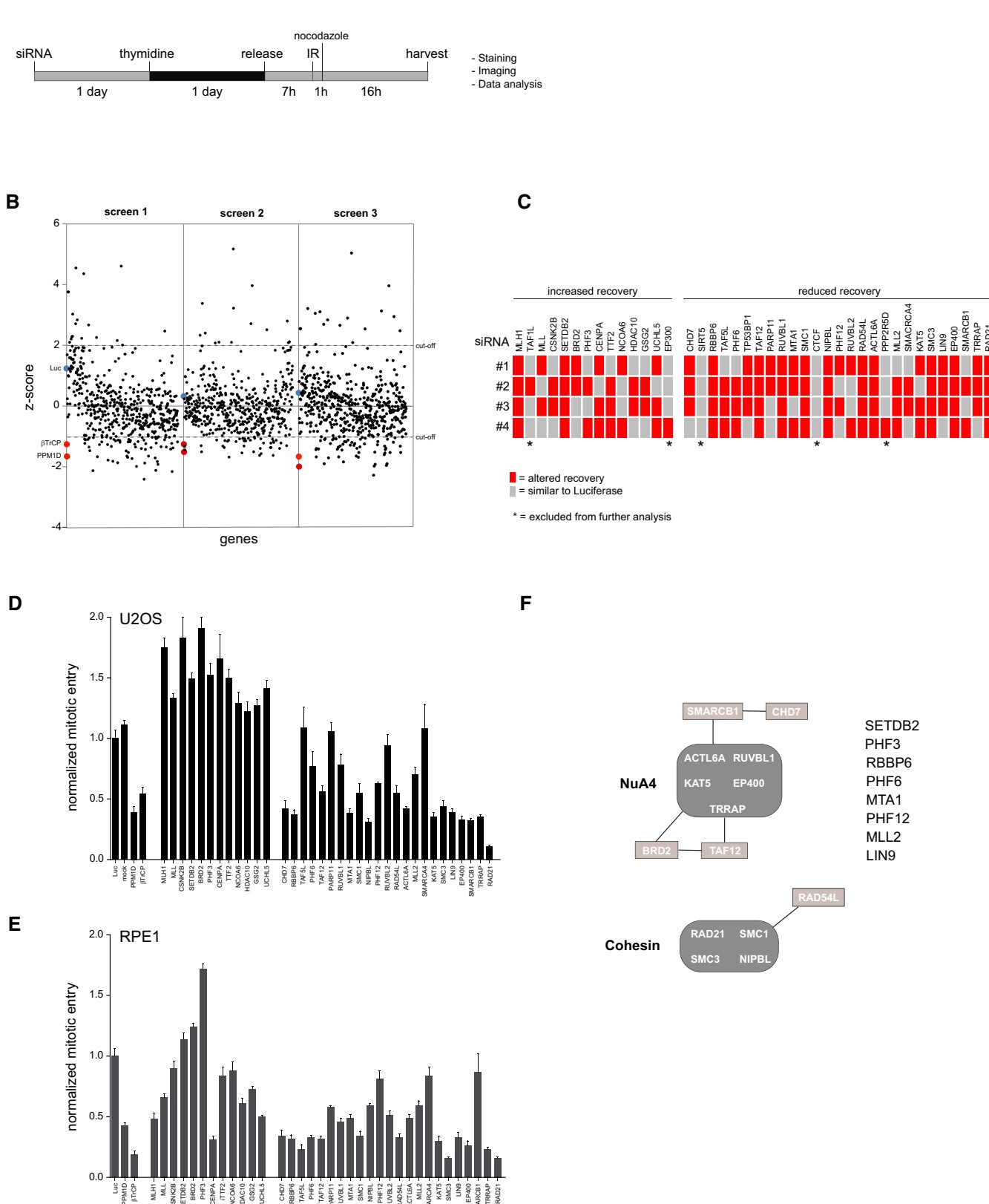

**Figure 1.**

We expected to identify chromatin regulators involved in different aspects of the DDR and recovery process as hits in our screen. We reasoned that the ability of previously damaged G2 cells to continue cell division is influenced by (i) recovery competence, which is the ability of cells to recover for some time through the p53-dependent transcriptional regulation of genes required for cell cycle restart like cyclins A, B, and CDK1 [16,35], (ii) ATM and ATR-mediated DNA damage-induced signaling, and (iii) the completion of DSB repair (Fig 2A). Therefore, we designed three different secondary assays to sub-classify the different candidates, allowing us to address how the identified hits regulate recovery from the DNA damage checkpoint. First, we measured their effect on checkpoint recovery competence in cells depleted for p53. Whereas transformed cells like U2OS maintain the competence to recover, primary RPE1 cells lose this ability rather quickly, which makes RPE1 the more representative cell model to study this cell fate-determining response [36,37]. While studying the recovery competence after co-depletion of p53 in RPE1 cells, we found that similar to the PPM1D-positive control, depletion of PHF3, RBBP6, RAD54L, KAT5, SMC3, PHF6, LIN9, TAF12, SMC1, NIPBL, ACTL6A, TRRAP, and RAD21 affected recovery in a p53-dependent manner, indicating the involvement of these genes in the regulation of DNA damage-induced p53 activation (Fig 2B). Next, we determined which genes disturb recovery by affecting the DNA damage checkpoint itself. For this, U2OS cells were treated with a high dose of IR (10 Gy) in order to prevent them from progressing into mitosis. Subsequently, cells were incubated with ATM/ATR inhibitors to override the DNA damage-induced checkpoint and induce recovery. We observed that inhibition of ATM and ATR activity in PHF3, CHD7, RBBP6, MTA1, SMC1, NIPBL, PHF12, RAD54L, MLL2, SMC3, LIN9, EP400, SMARCB1, TRRAP, and RAD21-depleted cells reversed the recovery defect in these cells (Fig 2C). This indicates that these genes modulate recovery through regulation of ATM/ATR-mediated signaling and that these hits are possibly involved in DNA damage checkpoint signaling and/or DNA repair.

The identified chromatin regulators were subsequently classified, thereby revealing an initial understanding of their potential mechanism of action. Cohesion-associated hits regulate recovery in both an ATM/ATR-dependent manner and a p53-dependent manner. This was less clear for the NuA4-related genes as we found that loss of CHD7, EP400, SMARCB1, and TRRAP regulates recovery only in an ATM/ATR-dependent manner, whereas TAF12, ACTL6A, KAT5, and TRRAP do so in a p53-dependent fashion in RPE1 cells. This indicates that subunits of this complex may have variable functions in response to DNA damage.

As the completion of repair is a prerequisite for cells to recover in the setting used for our screens, we finally investigated whether either of the hits was involved in DNA repair. Therefore, upon the depletion of individual candidates, we followed the induction and resolution of DSBs, measured by the presence of IR-induced foci (IRIF) of phosphorylated histone H2AX (γH2AX) and 53BP1 at 2 h after IR, as well as during recovery at 24 h after IR by immunofluorescence analysis. Loss of PHF6 and RAD21 led to increased γH2AX foci after IR, whereas SETDB2, BRD2, PHF3, RBBP6, TAF12, MLL2, KAT5, LIN9, EP400, SMARCB1, and TRRAP resulted in a decreased number of γH2AX foci (Figs 2D and EV1A). Downregulation of PHF3, RUVBL1, SMC1, and NIPBL caused more 53BP1 foci, whereas loss of RBBP6, PHF6, MTA1, PHF12, ACTL6A, and SMC3 resulted in a reduced number of 53BP1 IRIF (Figs 2E and EV1B). These results demonstrate that depletion of 20 of our hits affected the clearance of γH2AX and/or 53BP1 IRIF, suggesting that these genes are involved in DNA repair. Inefficient DNA repair often results in genome instability and thereby in a loss of viability. Therefore, we also investigated whether depletion of our hits influenced the survival of RPE1 cells upon IR. p53 was concomitantly depleted to minimize the influence of the G1 checkpoint on survival. Compared with the control, loss of TAF12, MTA1, PHF12, LIN9, and TRRAP resulted in increased cell survival, while depletion of BRD2, CHD7, RBBP6, PHF6, RUVBL1, NIPBL, ACTL6A, KAT5, EP400, SMARCB1, and RAD21 led to reduced viability after IR (Fig EV1C). As the majority of these genes were similar to the ones identified to be involved in the DDR (Fig 2C–E), these results combined suggest that they are important factors for genome stability maintenance and cell survival after damage, being one group actively required for DNA repair (low γH2AX and 53BP1 focus formation and lower survival than control cells) and another group required for regulating aspects of DNA repair (low γH2AX/53BP1 focus formation and higher survival). Combined, our siRNA-based primary and secondary screens led to the identification of 22 candidates among chromatin modifiers as novel regulators of recovery from the DNA damage checkpoint arrest.

Next, we sought to further validate PHF3, ACTL6A, and BRD2 as hits from our screen. Checkpoint recovery after the depletion of these proteins using four separate siRNA oligonucleotides was measured, and concurring knockdown efficiency was determined by Western blot. For PHF3, the knockdown by two out of four siRNA oligos led to increased checkpoint recovery. However, the effect did not correlate with the level of efficiency in protein depletion by the individual siRNAs. We found that recovery was

**Figure 2.  Classification of the screen hits.**

A       Overview of the cellular mechanisms that influence DNA damage-induced G2 checkpoint recovery.

B       RPE1 cells were depleted for p53 in combination with the indicated proteins by siRNA, in conditions used for the primary screen (2 Gy of IR). Mitotic entry was analyzed by MPM2 and PI staining by flow cytometry. For normalization, see Materials and Methods. Error bars represent the SEM of three independent experiments.

C       U2OS cells were transfected with the indicated siRNA oligonucleotides in a setup similar to the primary screen, but with 10 Gy of IR and in the presence of ATM and ATR inhibitors. Mitotic entry was analyzed by MPM2 and PI staining by flow cytometry. For normalization, see Materials and Methods. Error bars represent the SEM of three independent experiments.

D, E    U2OS cells were transfected with the indicated siRNA oligos using the protocol of the primary screen and fixed for immunofluorescence after 0, 2, and 24 h after 5 Gy IR. Samples were stained with antibodies for γH2AX (D) or 53BP1 (E) and analyzed through automated confocal microscopy and specifically developed image analysis software. From one experiment, two to four separate images with a total of at least 100 cells were analyzed for each knockdown. Shown is the number of γH2AX (D) and 53BP1 (E) foci per cell. In the box plots, the median with the first and third quartile is indicated, and the whiskers are drawn down to the 10th percentile and up to the 90th.

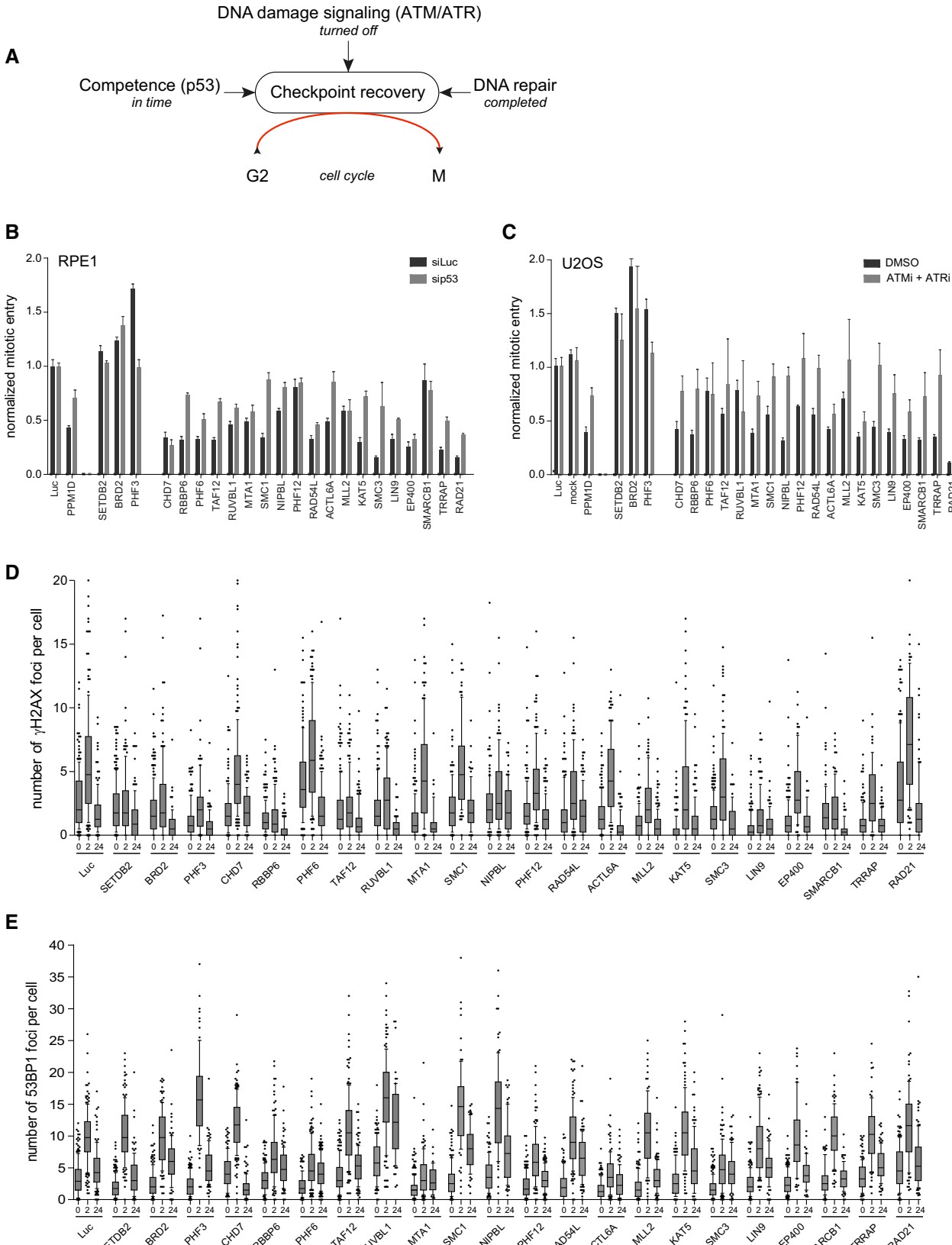

**Figure 2.**

decreased upon ACTL6A downregulation by all 4 siRNAs, correlating with the level of protein depletion. For BRD2, we observed increased recovery by 3 out of 4 siRNAs (Fig EV2A). To investigate whether PHF3, ACTL6A, and BRD2 were acting locally, at the site of the DNA lesion, their re-localization on laser-induced DNA breaks was investigated by immunofluorescence on fixed cells. In contrast to PHF3, ACTL6A and BRD2 were accumulating at laser-induced DNA lesions (Fig EV2B). The accumulation of BRD2 was confirmed by live cell imaging of YFP-BRD2, which showed that BRD2 accumulates as fast as NBS1 to damaged DNA at laser-stripes, but with a shorter retention time at DNA lesions as compared to that of NBS1 (Fig EV2C). This is in accordance with a recent study showing that BRD2 is recruited to nuclease-induced DSBs [38].

Interestingly, another hit from our screen, the transcriptional regulator PHF6 (PHD finger 6), was found to be mutated in Börjeson–Forssman–Lehmann syndrome, which is a rare X-linked genetic disorder characterized by mental retardation, craniopharyngeal abnormalities, and hematological cancers [39–44]. PHF6 was also recently identified as a potential sensitizing factor for drug treatment in resistant glioblastoma [45], which indicates that this protein is important for development and tumorigenesis. Moreover, PHF6 localizes within the nucleolus, where it is important for the repression of ribosomal DNA transcription [46,47]. Depletion of PHF6 was reported to lead to increased transcription of ribosomal DNA, resulting in a DNA damage-dependent stabilization of p53 and subsequent cell cycle arrest [46]. On that note, ribosomal RNA synthesis is inhibited after DNA damage, and this response is mediated by, besides PHF6, DNA-PK, and PARP1, factors promoting the repair of DSBs through NHEJ [48,49]. Mass spectrometry studies have indicated that PHF6 might be part of the NuRD complex, an ATP-dependent chromatin remodeler that also harbors histone deacetylation activity [42,43,50]. In addition to being involved in the regulation of transcription [43], the chromatin-remodeling activity of the NuRD complex has also been shown to be important for the DDR [51–53], in which the complex so far has been implicated in the repair of DSBs through HR [54–57]. Finally, PHF6 was reported to be a potential substrate of the kinases ATM/ATR and PLK1 [5,58], and such phosphorylation events may influence PHF6 cellular localization or the interaction with other proteins.

Encouraged by these findings, we sought to further study the role of PHF6 in checkpoint recovery. The observed effect of increased γH2AX and reduced 53BP1 IRIF (Fig 2D and E), as well as reduced viability after exposure to IR (Fig EV1C) upon knockdown of PHF6, suggested that PHF6 modulates IR-induced G2 checkpoint recovery through DNA repair and possibly p53 (Fig 3A). To investigate this hypothesis, the recovery potential was determined after downregulating PHF6 using four different siRNA oligonucleotides in RPE1 and U2OS cells (Figs 3B and EV3A). We observed that all four siRNA oligos partially reduced recovery in RPE1 cells, and the defect in recovery largely correlated with the knockdown efficiency, which was examined by Western blot (Fig 3C). Not all individual siRNA oligonucleotides caused a similar recovery defect in U2OS cells though, despite the fact that PHF6 protein depletion appeared to be equally efficient (Fig EV3B). A possible explanation for this variation is that PHF6 contains three and possibly more transcript variants, with cell type-specific functions [40,59]. Potential synergistic loss of p53 with knockdown of PHF6 using a particular siRNA oligonucleotide, as reported before for genes involved in DNA repair [60], could also result in a differential response.

To independently confirm the involvement of PHF6 in checkpoint recovery, we knocked out PHF6 in U2OS cells by CRISPR/Cas9-mediated genome editing (Fig EV3C). Knocking out PHF6 caused reduced checkpoint recovery which could be rescued significantly by exogenous expression of GFP-PHF6 in these cells. These findings clearly validate PHF6 as a bona fide hit in our screen and its specific involvement in checkpoint recovery (Fig 3D and E).

Next, we confirmed that the knockout (KO) of PHF6, similar to its knockdown (Fig 2D), led to reduced 53BP1 accumulation into DSB-containing foci at 1 hour after IR (Fig EV3D and E). Knockdown or knockout of PHF6 did not alter normal cell cycle distribution (Fig EV3F and G), indicating that the observed reduction in 53BP1 IRIF and recovery in those conditions are unlikely to be due to defects in cell cycle progression. We additionally observed that 53BP1 IRIF intensity was decreased in PHF6 knockout cells as compared to the wild-type cells (Fig 3F), implying that PHF6 is necessary for the recruitment of 53BP1 at DSBs. As seen after PHF6 depletion, knockout of PHF6 also resulted in an increase in γH2AX foci at early and late time points after IR, suggesting that not all breaks are efficiently repaired in cells lacking PHF6 (Figs 3G and EV3D). Comet assay analysis confirmed the presence of significantly more DNA breaks 1h after IR in cells lacking PHF6 in unperturbed conditions as compared to control cells, indicating that lesions are repaired less efficiently in the absence of PHF6 (Fig 3H). Altogether

**Figure 3. PHF6 regulates G2 checkpoint recovery.**

A    Graphical explanation of the effect of PHF6 on checkpoint recovery through DNA repair and p53.

B, C    U2OS cells were transfected with four different siRNA oligonucleotides targeting PHF6, synchronized in G2, and subsequent recovery after 5 Gy of IR and the addition of nocodazole were analyzed by flow cytometry with MPM2 staining (B), or cells were lysed and analyzed using Western blotting with the indicated antibodies (C). PPM1D and βTrCP were used as positive controls. Error bars represent the SEM of three independent experiments. Statistical significance was determined using a two-tailed, unpaired t-test (**P < 0.01, ***P < 0.001).

D, E    U2OS WT, PHF6 knockout, and PHF6 knockout cells that were reconstituted with GFP-PHF6 were synchronized in G2, and recovery was determined after 2 Gy of IR using flow cytometry with pHH3 staining (D), or cells were lysed and analyzed using Western blotting with the indicated antibodies (E). Error bars represent the SEM of three independent experiments. Statistical significance was determined using a two-tailed, unpaired t-test (**P < 0.01). The remaining PHF6 signal in the PHF6 KO sample (E) is likely due to a crossreaction at similar height as PHF6 (also see Fig EV3C).

F, G    U2OS WT and PHF6 knockout cells were irradiated using 3 Gy of IR, and cells were fixed at different time points for immunofluorescence. In (F) is presented 53BP1 IRIF intensity. In (G) is shown the number of γH2AX foci per cell. Error bars represent the SD of three independent experiments. Statistical significance was determined using a two-tailed, unpaired t-test (****P < 0.0001).

H    U2OS WT and PHF6 knockout cells were left untreated or irradiated with 3 Gy and processed after 1 h for comet assay analysis. The comet tail moment was analyzed in at least 50 individual cells. Error bars represent the SD of three independent experiments. Statistical significance was determined using a two-tailed, unpaired t-test (*P < 0.05, ****P < 0.0001).

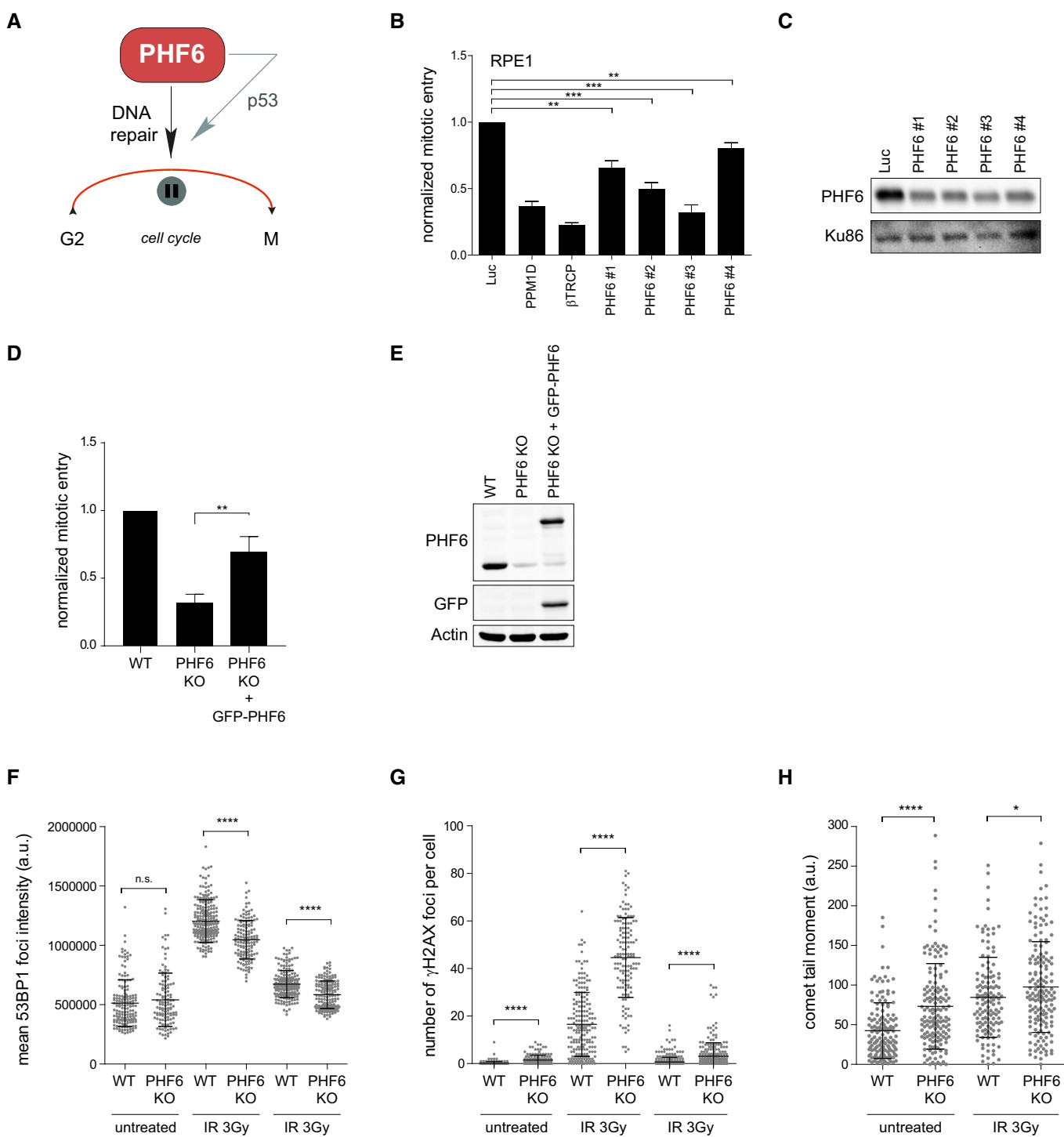

**Figure 3.**

these data pinpoint toward a role of PHF6 in efficient checkpoint recovery after IR by promoting DSB repair.

PHF6 contains two PHD domains (PHD1 and PHD2), zinc finger-like motifs known to be involved in chromatin regulation [61]. The PHD1 domain was suggested to interact with upstream binding factor (UBF), a ribosomal DNA transcription factor [46,47,62]. The PHD2 domain, on the other hand, was reported to bind double-stranded

DNA *in vitro*, and similar domains have been found in chromatin-associated proteins [63]. To study the involvement of the PHD domains of PHF6 in the DDR and recovery, U2OS PHF6 KO cells were complemented with either wild-type GFP-PHF6 or mutant versions in which either the PHD1 domain or the PHD2 domain was deleted (Fig EV4A and B), after which IR-induced γH2AX and 53BP1 focus formation was determined. Complementation using wild-type PHF6 and the PHD1

and PHD2 domain mutants was able to partially rescue the increase in IR-induced γH2AX foci observed in PHF6 KO cells (Fig EV4C). However, although the expression of PHF6 wild-type rescued the decrease in the number of 53BP1 foci upon IR, the PHD deletion mutants did not (Fig EV4D), indicating that both PHD domains are required for 53BP1 focus formation in response to DSBs. The resolution of IR-induced γH2AX foci was also dependent upon both PHD domains and correlated with the inability of both mutants to rescue G2 checkpoint recovery after DNA damage (Fig EV4E and F). Together, these results suggest that PHF6, through its PHD domains, may bind and/or modulate chromatin in response to damage, allowing for efficient DNA repair and recovery from DNA damage.

The fact that the accumulation of 53BP1, implicated in DSB repair [27], into IRIF is diminished in PHF6 knockout cells prompted us to study whether PHF6 is directly involved in the repair of DNA DSBs. Interestingly, investigating the recruitment of PHF6 to laser-induced damage by direct fluorescence in living cells demonstrated that GFP-PHF6 is rapidly and transiently recruited to laser-induced DNA breaks (Fig 4A). This recruitment, as well as that of NBS1, which served as a positive control, was unaffected by incubating cells with ATM and ATR inhibitors (Fig EV5A and B). In contrast, inhibition of PARP1/2 by Olaparib completely blocked the accumulation of PHF6 at DNA breaks (Fig 4B). Recent data indicate a role for PARP1 in NHEJ by promoting the recruitment of NHEJ factors to DNA ends [49]. Together with the effect of PHF6 on the recruitment of 53BP1, our data suggest that PHF6 might play a direct role in the initiation of DNA repair, possibly in NHEJ, acting downstream of PARP1. To verify this hypothesis, we employed GC92 and DR-GFP reporter assays, which allowed us to monitor the repair of an I-SceI nuclease-induced DSB by NHEJ or HR, respectively [64–66]. Importantly, and in accordance with the effect on 53BP1 IRIF, loss of PHF6 inhibits repair through end joining (Fig 4C). Besides, PHF6 depletion concomitantly resulted in increased HR with 2 out of 3 siRNA oligonucleotides, whereas a reduction in HR was observed with the additional siRNA oligonucleotide (Fig EV5C). siRNA oligonucleotides have been shown to reduce HR through off-target effects particularly on RAD51 expression [67]. Combined with our observation that in U2OS the siRNA oligonucleotides targeting PHF6 sometimes showed a differential response compared with RPE1 cells, this could provide an explanation for this observed difference.

We considered it a possibility that PHF6 controls NHEJ by transcriptional regulation of factor(s) regulating this DSB repair pathway. We therefore examined the expression of a broad range of NHEJ-associated factors by Western blot after PHF6 depletion by siRNA

and in the PHF6 KO cells (Fig EV5D). No differences in proteins levels of any of the factors were observed, suggesting that PHF6 controls NHEJ more directly. Furthermore, we found that knockout of PHF6 resulted in a similar sensitivity of U2OS cells to IR after depletion of core-NHEJ factor XRCC4. Moreover, downregulation of XRCC4 in PHF6 KO cells did not result in increased IR sensitivity as compared to control depletion (Figs 4D and EV5E), corroborating a NHEJ repair defect upon PHF6 loss and suggesting that PHF6 functions in the same pathway of classical NHEJ as XRCC4.

Finally, to confirm that PHF6 is indeed involved in classical, rather than alternative, NHEJ [68], we depleted PHF6 from cells containing the GC92 reporter for NHEJ and examined repair of the break sites by sequence analysis of the repair junctions. We found that depletion of PHF6, similar to Ku80 knockdown, resulted in a decrease in direct re-ligation events, concomitantly with an increase in deletions, insertions, and microhomology usage at the break site, indicating a shift to alternative NHEJ and substantiating a role for PHF6 in classical NHEJ (Figs 4E and EV5F and G).

Altogether, our data identify PHF6 as a novel factor in the regulation of DNA repair through classical end joining. Inefficient repair by loss of PHF6 results in unrepaired DSBs and therefore a reduced recovery from the G2 checkpoint after IR (Fig 4F). In an earlier screening for factors controlling checkpoint recovery, Bruinsma *et al* [69] identified TLK2 as a novel regulator of checkpoint recovery, by regulating histone chaperone ASF1a. Interestingly, ASF1a was recently reported to promote NHEJ [70].

While our experiments clearly indicate a direct role for PHF6 in facilitating NHEJ, resolving the exact molecular mechanism by which PHF6 supports this effect requires further studies. PHF6 may influence the HDAC activity of the NuRD complex, as HDAC1 and HDAC2 have been implicated in NHEJ [71]. Alternatively, given its effect on 53BP1 focus formation, PHF6 might affect DSB resection, by inhibiting the (further) processing of DSBs, thereby promoting DNA repair by NHEJ.

Börjeson–Forssman–Lehmann syndrome and cancer-associated PHF6 mutations could potentially enable cells to deal differentially with DNA damage, with possible hazardous consequences for the stability of the genome. In addition, given our results demonstrating the effect of PARP1/2 on the localization of PHF6 at sites of DNA lesions, (co)treatment with PARP1/2 inhibitor would not be beneficial for the therapy of T-ALL/AML harboring PHF6 loss-of-function mutations. A better understanding of whether PHF6 mutations identified in tumors influence repair of DNA damage will be important to further resolve the clinical implications of our findings.

---

**Figure 4. PHF6 affects recovery by controlling DNA repair via classical NHEJ.**

A, B   U2OS cells stably expressing GFP-PHF6 and mCherry-NBS1 were laser-irradiated, and cells were analyzed by time-lapse imaging. In (B), the PARP inhibitor Olaparib was added 30 min before laser-induced damage. At least 60 cells were analyzed in three individual experiments, and error bars represent the SD (lower panel). Scale bar is 5 μm.

C   GC92 SV40 immortalized human fibroblasts containing the I-SceI NHEJ reporter were transfected with the indicated siRNA oligos and I-SceI. 48 h later, the cells were fixed and analyzed by flow cytometry. Represented is the relative repair efficiency as compared to the Luciferase control. Error bars represent the SEM of three independent experiments. Statistical significance was determined using a two-tailed, unpaired *t*-test (**$P < 0.01$, ***$P < 0.001$).

D   Clonogenic survival assays of U2OS WT and PHF6 knockout cells that were depleted for luciferase or XRCC4 and treated with 2, 3, or 4 Gy IR. Shown is the relative survival as compared to the undamaged control. Error bars represent the SEM of three individual experiments.

E   GC92 SV40 immortalized human fibroblasts containing the I-SceI NHEJ reporter were transfected with the indicated siRNA oligos and I-SceI. 48 h later, genomic DNA was extracted and repair of I-SceI cut sites analyzed through junction analysis (*n* = independently derived sequences, see Materials and Methods for details).

F   Model showing early recruitment of PHF6 to DNA DSBs in a PARP1/2-dependent manner, possibly to remodel the chromatin for efficient DNA repair through classical NHEJ and thereby to promote recovery from the G2 checkpoint arrest.

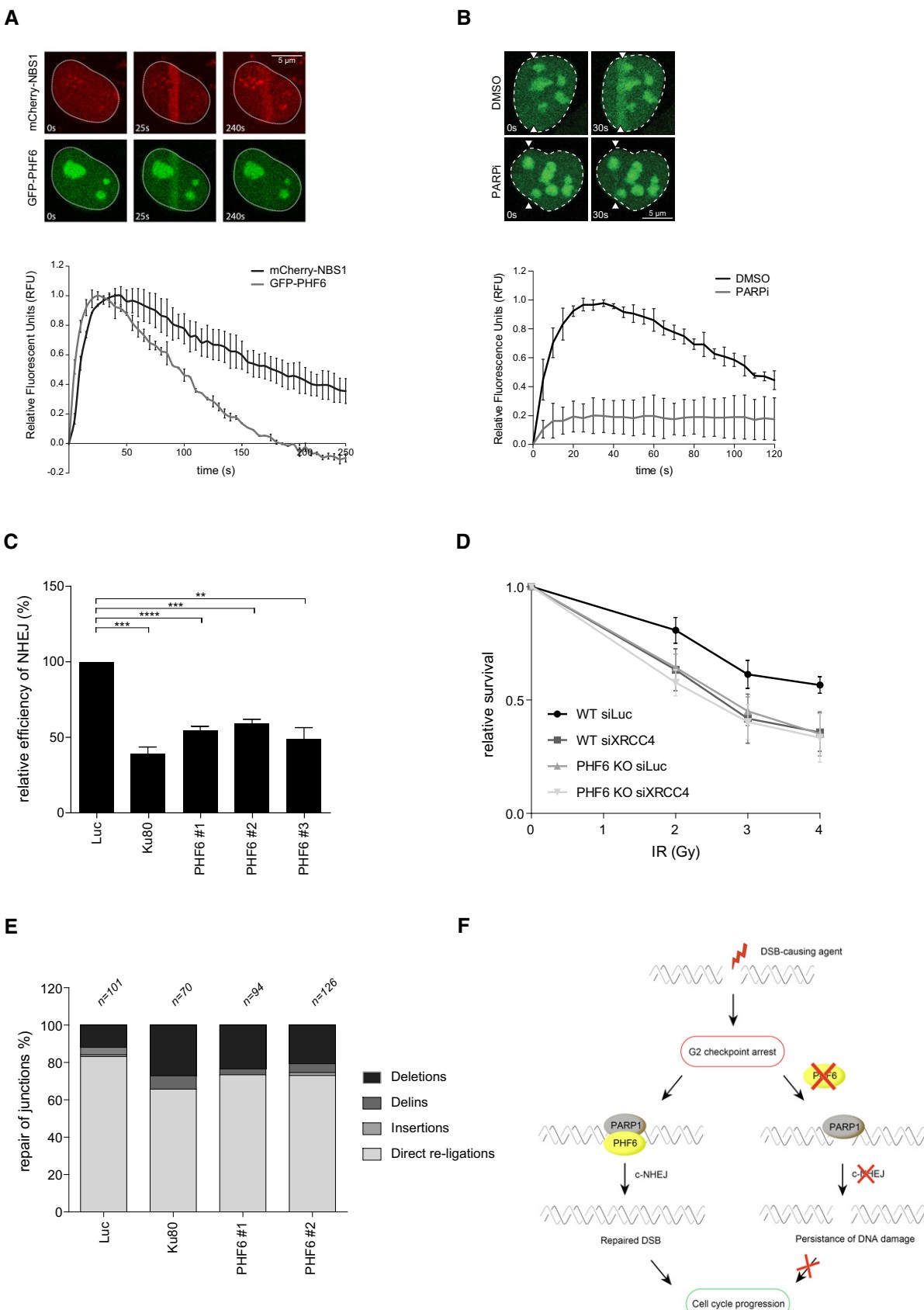

**Figure 4.**

# Materials and Methods

## siRNA screen

The screen was performed with the Dharmacon ON-TARGETplus siRNA chromatin library (custom-made, order numbers 245120-245146, 245170, and 246203), covering 529 genes with four siRNA oligos per gene [25]. For the deconvolution screen, the four single siRNAs of the SMARTpool were tested separately. The primary screen was performed in triplicate, and the deconvolution screen was performed in duplicate. The siRNA libraries were aliquoted in 96-well plates to a final concentration of 20 nM of siRNA per well. To each plate were added oligonucleotides targeting luciferase, PPM1D, and βTrCP as controls. One thousand cells were added to each well after incubation of RNAiMax (Life Technologies) transfection reagents according to the manufacturer's guidelines.

Cells were fixed using 4% formaldehyde for 10 min and washed using PBS before staining using primary and secondary antibodies. Cells were analyzed using a Cellomics ArrayScan VTI (Thermo Scientific) using a 10× (0.50 NA) objective. DAPI staining was used to identify the cells, which were scored to be mitotic if the phospho-histone H3 (pHH3) signal reached a set threshold. Four images were acquired per well. Image analysis was performed using Cellomics Target Activation BioApplication (Thermo Scientific).

## siRNA and CRISPR/Cas9

ON-TARGETplus siRNAs (SMARTpool or single oligos) targeting luciferase, TP53, Artemis, PPM1D, βTRCP, KU80, CtIP, XRCC4, and the custom-ordered library (order number 431436) containing SETDB2, BRD2, PHF3, CHD7, RBBP6, PHF6, TAF12, RUVBL1, MTA1, SMC1, NIPBL, PHF12, RAD54L, ACTL6A, MLL2, KAT5, SMC3, LIN9, EP400, SMARCB1, TRRAP, and RAD21 (Table EV2) were purchased from Dharmacon and were transfected as described above. All siRNA transfections were performed at least 24 h before the start of the experiment.

PHF6 knockout U2OS cell lines were generated using CRISPR/Cas9 genome editing. CRISPR sequences were designed targeting PHF6 (exon 2 guide: AATATCTGAAAACCAGAAGG and exon 4 guide: CATTGTCCTGGAGCAACAAT) and cloned into pX458. Cells were transfected with the targeting constructs in addition to a plasmid containing a guide RNA to the zebrafish TIA gene (GGTATGTCGG GAACCTCTCC) and a P2A sequence followed by a puromycin resistance gene, flanked by two TIA target sites. Co-transfection resulted in infrequent integration of the P2A-puromycin cassette at the targeted location, as previously described [72]. Successful in-frame integration of the gene rendered cells resistant to puromycin. Two days following transfection, the culture medium was supplemented with puromycin (2 μg/ml). Surviving colonies were clonally expanded and screened by Western blot for expression of PHF6.

## PHF6 mutants

Different mutations were introduced in the GFP-PHF6 plasmid using the QuikChange Site-Directed Mutagenesis Kit (Agilent Technologies) to obtain the following constructs: ΔPHD1 mutation (deletion of amino acids 17–131) and ΔPHD2 mutation (deletion of amino acids 212–329).

## Cell culture, irradiation, and inhibitors

U2OS cells were cultured in Dulbecco's modified Eagle's medium (DMEM), and RPE-1 hTert cells were maintained in Dulbecco's Modified Eagle Medium/Nutrient Mixture F-12 (DMEM/F12, Gibco). Both were supplemented with penicillin/streptomycin and 10% tetracycline-approved fetal calf or bovine serum. G2 synchronization was performed by treating cells with 2.5 mM thymidine (Sigma) for 24 h, after which cells were released by washing twice with PBS. Cells were irradiated at 7–8 h after release. When indicated, G2-synchronized U2OS cells were treated with the respective inhibitors for one hour before irradiation and treated with 100 ng/ml nocodazole (Sigma) after irradiation.

Irradiation was performed using a Gammacell Exactor (Best Theratronics) with a $^{137}$Cs source, a CellRad (Faxitron), or a WOMed X-Ray Therapy Unit T-105. Etoposide (Sigma) was used in a final concentration of 3 μM.

ATM inhibitor (KU55933) from Calbiochem, ATR inhibitor (VE-821) from Axon Medchem, and PARP1/2 inhibitor (Olaparib) from Sigma were used in a final concentration of 10 μM.

## Viability assay and clonogenic survival

One hundred cells were seeded into 96-well dishes and left for 16 h to attach. Cells were left untreated or irradiated (4 Gy) and subsequently incubated for 10 days. Plates were fixed using methanol, and surviving cells were stained with crystal violet. Optical density was measured using a plate reader. For each knockdown, the irradiated sample was divided by the untreated and all samples were normalized to the luciferase control, which was set to 1.

For clonogenic survival, 250 cells were seeded into 6-well dishes and then treated with the indicated doses of IR. Following 7–10 days in culture, cells were fixed, stained, and colonies were counted. Triplicate cultures were scored for each treatment. Shown is the relative survival as compared to the undamaged control, and the error bars present the standard error of the mean of three independent experiments.

## Antibodies

53BP1 (H-300), Actin (I-19), GAPHD (FL-335), PARP1 (F-2), Ku70 (C-19), Ku86 (C-20), and DNA-PKcs (H-163) were purchased from Santa Cruz Biotechnology; 53BP1 (ab172580), Ligase IV (ab193353), Nbs1 (ab175800), Histone H3 (ab1791), and SMC1 (ab9262) were purchased from Abcam; pSer139 H2AX (clone: JWB301), pS10 Histone H3, and phospho-Ser/Thr-Pro MPM-2 were purchased from Millipore; PHF6 (68262), XLF (2258), and Artemis (183) were purchased from Novus; XRCC4 (40455) was purchased from Signalway Antibody, and CtIP was purchased from Active Motif. Anti-TopBP1 was described before [73]. Antibodies against human ACTL6A, PHF3, BRD2, Mre11, Rad50, and RIF1 were raised in rabbits. For this, the cDNA corresponding the amino acids 1–360 of ACTL6A, 90–400 of PHF3, 450–750 of BRD2, 182–480 of Mre11, 200–500 of Rad50, and 1,200–1,500 of RIF1 was cloned into pET30a vector (Novagen) for expression in *Escherichia coli* as a His fusion protein. Fragments were purified using Ni-NTA (Qiagen) following the manufacturer's instructions, and rabbits were immunized. Serum was collected and purified against the corresponding antigen

as described [74]. HRP-coupled secondary antibodies used for Western blot were purchased from DAKO. For immunofluorescence, Alexa-coupled secondary antibodies were purchased from Molecular Probes.

## Immunofluorescence

Cells were grown on coverslips and fixed with 2% PFA for 20 min, permeabilized using 0.5% Triton X-100 for 5 min, and blocked in 2% BSA for 1 h. Samples were incubated with primary antibodies o/n at 4°C. After washing, cells were incubated with secondary antibodies and DAPI for 1h at RT. Coverslips were mounted onto glass slides using ProLong (Life Technologies). Pre-extraction was performed by incubating cells with 0.5% Triton X-100 for 2 min before fixation.

Images were taken using a Leica SP5 confocal microscope equipped with a 63× NA 1.40 oil immersion objective and an Argon laser and 405 nm, 561 nm, 633 nm diode lasers, or a Zeiss Cell Observer fluorescent microscope equipped with a 63× NA 1.3 water immersion objective and ZEN imaging software. A number of IRIF and fluorescence intensity were evaluated in ImageJ (NIH).

## COMET assay

Neutral Single Cell Gel Electrophoresis (SCGE) was carried out using the CometAssay® ES II kit (Trevigen) according to the manufacturer's instructions. Images were taken using a Zeiss Cell Observer fluorescent microscope, and the tail moment of at least 50 cells per experiment was analyzed with the TriTek CometScore software.

## Automated 53BP1 and γH2AX IRIF analysis

Images were taken using a Leica SP5 confocal microscope equipped with a 40× NA 1.40 water immersion objective and an Argon laser (at 488 nm), and 405 nm, 561 nm, and 633 nm diode lasers. 53BP1 and γH2AX ionizing radiation-induced foci (IRIF) were analyzed in U2OS cells 0, 2, and 24 h after 5 Gy. IRIF were evaluated in ImageJ, using a custom-built macro that enabled automatic and objective analysis of the foci. Cell nuclei were detected by thresholding the (median-filtered) DAPI signal, after which touching nuclei were separated by a watershed operation. Segmentation mistakes were corrected manually. After maximum intensity projection, the foci signal was background-subtracted using a Difference-of-Gaussians filter. For every nucleus, foci were identified as regions of adjacent pixels satisfying the following criteria: (i) The gray value exceeds the nuclear background signal by a set number of times (typically 2–4×) the median background standard deviation of all nuclei in the image and is higher than a user-defined absolute minimum value; (ii) the area is larger than a defined area (typically two pixels). These parameters were optimized for every experiment by manually comparing the detected foci with the original signal.

## Laser micro-irradiation

Multiphoton laser micro-irradiation was essentially performed as described previously [75]. Cells, grown on coverslips, were placed in a Chamlide CMB magnetic chamber, and the medium was replaced by $CO_2$-independent Leibovitz's L15 medium supplemented with 10% FCS and penicillin-streptomycin. Laser micro-irradiation was carried out on a Leica SP5 confocal microscope equipped with an environmental chamber set to 37°C. DSB-containing tracks (1.5 μm width) were generated with a Mira mode-locked titanium-sapphire (Ti:Sapphire) laser ($\lambda = 800$ nm, pulse length = 200 fs, repetition rate = 76 MHz, output power = 80 mW) using a UV-transmitting 63× 1.4 NA oil immersion objective (HCX PL APO; Leica). Confocal images were recorded before and after laser irradiation at 5-or 10-s time intervals over a period of 5–10 min. The protocol for fixed cells was as previously described [76]. In brief, cells were grown on coverslips and incubated with Hoechst before micro-irradiation.

## Flow cytometry

For cell cycle analysis, cells were fixed in 70% ethanol at 4°C o/n. After fixation, cells were washed with PBS, and the DNA was stained with propidium iodide (PI). Samples were analyzed by Macsquant Analyzer (Miltenyi) and Flowlogic software. G2 checkpoint analysis was performed as described above, but cells were stained with antibodies against MPM2 or pHH3 to determine the number of mitotic cells. At least 15,000 cells were analyzed per condition, and three independent experiments were performed using a FACS Calibur (BD Biosciences) or a Macsquant Analyzer (Miltenyi) and analyzed using CellQuest or Macsquantify software, respectively. The number of mitotic cells after IR was divided by the number of mitotic cells in untreated conditions resulting in the relative mitotic entry (RME). For each experiment, the RME was normalized to the siLuciferase control, which was set to 1.

## HR and NHEJ assay

GC92 fibroblasts and DR-GFP reporter U2OS cells were used to measure the repair of I-SceI-induced DSBs by NHEJ and HR, respectively [64–66]. Briefly, 48 h after siRNA transfection, cells were transfected with the I-SceI expression vector and mCherry. The fraction of CD4 (NHEJ) or GFP (HR)-positive cells among the mCherry-positive cells was measured 48 or 72 h later by FACS on a LSRII flow cytometer (BD Bioscience) using FACSDiva software. Quantifications were performed using Flowing Software.

Non-homologous end joining junctions were analyzed in GC92 fibroblasts as described in Ref. [77]. Briefly, GC92 fibroblasts containing the NHEJ reporter were transfected with siRNA oligonucleotide against the indicated genes. After 48 h, I-SceI was expressed, and 48 h later, cells were lysed and genomic DNA was extracted, followed by PCR and cloning of the amplicons in pGEM T-easy. DNA was transformed, and > 70 single clones were analyzed by Sander-sequencing. In-house software was used to analyze the obtained sequences [78]. Independently derived sequences were classified into four distinct categories: (i) simple deletions (deletions), (ii) deletions accompanied by an inclusion of DNA (delins), (iii) insertions of DNA at the break site (insertions), and iv) sequences that did not contain a mutation in the amplified targeted region (direct re-ligations). The sequences containing deletions were further categorized into different bins for deletion size and microhomology usage.

## Western blotting

For whole cell extracts, cells were lysed in Laemmli sample buffer. Protein concentration was determined by BCA protein assay (Thermo Scientific), and samples were analyzed by Western blot.

## Statistical analysis

Statistical significance in all experiments was determined using a two-tailed, unpaired *t*-test (*$P < 0.05$, **$P < 0.01$, ***$P < 0.001$, ****$P < 0.0001$).

**Expanded View** for this article is available online.

## Acknowledgements
We thank Jozef Gécz for sharing GFP-PHF6, Susan Janicki for YFP-BRD2, Thijn Brummelkamp for the pTIA-Puro vector, Feng Zhang for pX458 (Addgene plasmid #48138), and Bernard Lopes for the GC92 fibroblast cells. We thank the iPSC CRISPR Facility from the University Medical Center Groningen and ERIBA for assistance with the generation of PHF6 knockout cell lines, David Egan from the Cell Screening Core at the Department of Cell Biology from the UMC Utrecht with technical assistance for screening, Govind Pai from Oncogenetics at the Amsterdam UMC for technical assistance with flow cytometry, the Bioimaging, Flow Cytometry facilities at the Netherlands Cancer Institute for their support, and Antonio Catalán, Luciano Benítez, and Pilar Vaswani (Servicio de Física Médica) from the Hospital Universitario de Canarias for use of the WOmed X-Ray Therapy Unit. We thank Louis Vermeulen for advice and critical reading of the manuscript. This work was supported by the Cancer Center Amsterdam (DOW, RMFW), Ministerio de Ciencia, Innovación y Universidades (SAF2016-80626-R to RF and VAJS and BFU2017-90889-REDT to VAJS), co-funded by EU-ERDF. IAV is supported by a predoctoral fellowship from the Gobierno de Canarias and HvA by an ERC Consolidator grant from the European Research Council and a VICI grant from the Netherlands Organization for Scientific Research.

## Author contributions
DOW, RHM, and VAJS conceived the study and together with HA and RMFW designed the experiments. DOW and VAJS wrote the paper. DOW, IAV, WWW, BB, MBR, and VAJS performed the experiments and analyzed the data. RF generated the polyclonal antibodies.

## Conflict of interest
The authors declare that they have no conflict of interest.

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
