## [Review Process File · EMBO Reports]

PHF6 promotes non-homologous end joining and G2 checkpoint recovery

Daniël O. Warmerdam, Ignacio Alonso-de Vega, Wouter W. Wiegant, Bram van den Broek, Magdalena B. Rother, Rob M.F. Wolthuis, Raimundo Freire, Haico van Attikum, René H. Medema, Veronique A.J. Smits

Review timeline:

Submission date:	10 May 2019
Editorial Decision:	7 June 2019
Revision received:	27 September 2019
Editorial Decision:	28 October 2019
Revision received:	4 November 2019
Accepted:	5 November 2019

Editor: Esther Schnapp

Transaction Report:

1st Editorial Decision

7 June 2019

Thank you for the submission of your manuscript to EMBO reports. We have now received the full set of referee reports as well as referee cross-comments that are all pasted below.

As you will see, the referees acknowledge that the findings are potentially interesting. However, both referees 1 and 2 point out that it remains unclear what the molecular and functional role of PHF6 in the DDR is, and that the study should be further developed in this regard. All referees have different suggestions for how the study could be strengthened, and I would like to know whether you think that you can address them all. If this is not the case, we can discuss the detailed revisions further. I think the cross-comments are quite informative.

I would thus like to invite you to revise your manuscript with the understanding that the referee concerns must be fully addressed and their suggestions taken on board. Please address all referee concerns in a complete point-by-point response. Acceptance of the manuscript will depend on a positive outcome of a second round of review. It is EMBO reports policy to allow a single round of revision only and acceptance or rejection of the manuscript will therefore depend on the completeness of your responses included in the next, final version of the manuscript.

Revised manuscripts should be submitted within three months of a request for revision; they will otherwise be treated as new submissions. Please contact us if a 3-months time frame is not sufficient for the revisions so that we can discuss this further. You can either publish the study as a short report or as a full article. For short reports, the revised manuscript should not exceed 27,000 characters (including spaces but excluding materials & methods and references) and 5 main plus 5 expanded view figures. The results and discussion sections must further be combined, which will help to shorten the manuscript text by eliminating some redundancy that is inevitable when discussing the same experiments twice. For a normal article there are no length limitations, but it should have more than 5 main figures and the results and discussion sections must be separate. In both cases, the entire materials and methods must be included in the main manuscript file.

Regarding data quantification, please specify the number "n" for how many independent experiments were performed, the bars and error bars (e.g. SEM, SD) and the test used to calculate p-values in the respective figure legends. This information must be provided in the figure legends. Please also include scale bars in all microscopy images.

- 1) a .docx formatted version of the manuscript text (including legends for main figures, EV figures and tables). Please make sure that the changes are highlighted to be clearly visible.
- 2) editable tiff, jpg or eps-formatted figure files in high resolution (one file per figure). In order to avoid delays later in the process, please read our figure guidelines before preparing your manuscript figures at: http://www.embopress.org/sites/default/files/EMBOPress_Figure_Guidelines_061115.pdf
- 3) We replaced Supplementary Information with Expanded View (EV) Figures and Tables that are collapsible/expandable online. A maximum of 5 EV Figures can be typeset. EV Figures should be cited as 'Figure EV1, Figure EV2' etc... in the text and their respective legends should be included in the main text after the legends of regular figures.
 - For the figures that you do NOT wish to display as Expanded View figures, they should be bundled together with their legends in a single PDF file called *Appendix*, which should start with a short Table of Content. Appendix figures should be referred to in the main text as: "Appendix Figure S1, Appendix Figure S2" etc. See detailed instructions regarding expanded view here: <http://msb.embopress.org/authorguide#expandedview>.
 - Additional Tables/Datasets should be labeled and referred to as Table EV1, Dataset EV1, etc. Legends have to be provided in a separate tab in case of .xls files. Alternatively, the legend can be supplied as a separate text file (README) and zipped together with the Table/Dataset file.
- 4) a .docx formatted letter INCLUDING the reviewers' reports and your detailed point-by-point responses to their comments. As part of the EMBO Press transparent editorial process, the point-by-point response is part of the Review Process File (RPF), which will be published alongside your paper.
- 5) a complete author checklist, which you can download from our author guidelines [<http://embor.embopress.org/authorguide#revision>]. Please insert information in the checklist that is also reflected in the manuscript. The completed author checklist will also be part of the RPF.
- 6) Please note that all corresponding authors are required to supply an ORCID ID for their name upon submission of a revised manuscript.
- 7) Before submitting your revision, primary datasets produced in this study need to be deposited in an appropriate public database (see <http://msb.embopress.org/authorguide#dataavailability>). Please remember to provide a reviewer password if the datasets are not yet public. The accession numbers and database should be listed in a formal "Data Availability " section placed after Materials & Method (see also <http://msb.embopress.org/authorguide#dataavailability>). Please note that the Data Availability Section is restricted to new primary data that are part of this study.
 - * Note - All links should resolve to a page where the data can be accessed. *
- 8) We would also encourage you to include the source data for figure panels that show essential data. Numerical data should be provided as individual .xls or .csv files (including a tab describing the data). For blots or microscopy, uncropped images should be submitted (using a zip archive if multiple images need to be supplied for one panel). Additional information on source data and instruction on how to label the files are available at <http://msb.embopress.org/authorguide#sourcedata>.

We would also welcome the submission of cover suggestions, or motifs to be used by our Graphics

Illustrator in designing a cover.

I look forward to seeing a revised version of your manuscript when it is ready. Please let me know if you have questions or comments regarding the revision.

REFeree REPORTS

Referee #1:

Overview: Warmerdam et al. present a progressive genetic screening analysis to identify unappreciated factors required for recovery from the DNA damage G2 checkpoint. Not surprisingly, many screen hits were involved in DNA repair as their deficiency halts DNA damage checkpoint recovery due to deficient lesion clearance. The primary lead from this analysis was the implication of PHF6 in ensuring checkpoint recovery and DNA repair following DNA damage.

Novelty/Quality: The presented manuscript is well written by experts within the DDR/checkpoint field. Overall, it is of good quality with several complementary approaches (PHF6 loss-of-function addressed with multiple siRNAs and CRISPR) and examines DDR/checkpoint signaling from a less appreciated viewpoint of recovery. Further, this manuscript indicates the disease relevant factor PHF6 as a DDR factor. However, though generally of solid quality, the current manuscript version is highly descriptive and offers little insight into the molecular function and mechanism of PHF6 in the DDR response.

Major Points

- 1) The authors primary supposition is that loss of PHF6 leads to a repair defect following DNA damage as indicated by the elevated levels of the marker phosphorylated-H2AX. To properly support this claim, they should include neutral COMET assay analysis to better reflect the extent of damage still present in the cells at this time.
- 2) The authors should provide functional-mechanistic explanations for the role of PHF6. The genetic KO system is optimally suited to dissect the role of PHF6 through complementation, preferably also with disease variants.
- 3) Such complementation should address major DDR aspects for major PHF6 functional mutants (PHF6 recruitment, checkpoint recovery and death, DNA damage accumulation by COMET assay, NHEJ and 53BP1 pathway function, DNA end resection performance).

Minor Points

- figure 4 figure legend indicates panel F, need to change to E
- authors need to indicate radiation dosages used for the screens. Starting with figure 1A.

Referee #2:

The manuscript from Smits and colleagues presents results from a screen to identify chromatin proteins involved in recovery from the G2 checkpoint. Using a strategy the authors have exploited previously, they performed an siRNA screen on cells treated with IR and trapped in nocodazole, identifying numerous proteins that enhance or inhibit mitotic entry. These are validated in multiple cell lines to identify consistent hits and then sub analyzed as to whether they influence p53 activation or DNA damage signaling via ATM/ATR. They focus on several novel hits that include

BRD2, ACTL6A and PHF6. They show that PHF6 depletion or knockout impairs mitotic entry after DNA damage, reduces 53BP1 foci and impairs NHEJ and that PHF6 localizes directly to DSBs.

Overall the manuscript is well presented and data of high quality. The results are interesting and novel, as to my knowledge PHF6 has not been implicated in DNA repair previously and the screening results may be of interest to many labs. My only criticism would be that the role of PHF6 in NHEJ remains unclear. While data presented suggests a potential direct role, its other functions are not ruled out (ex. transcription of NHEJ genes, as has been shown for BRD2) and no data demonstrates that the endogenous protein is actually recruited to DNA breaks.

Main points

1. The manuscript culminates in a focus on PHF6 from Figure 3 on. In the PHF6 knockdown or knockout cells, is there a defect in cell cycle progression in untreated cells? Does this influence the appearance of 53BP1 foci that are cell cycle dependent?
2. Cell cycle status also influences NHEJ, it would be important to show data regarding the effects of PHF6 loss in order to properly evaluate the data in figure 3 and 4.
3. Why is co-localization with NBS1 investigated vs factors involved in NHEJ?
4. Does treating PHF6 KOs with DNA-PKcs inhibitor influence sensitivity or is it epistatic?
5. Are NHEJ proteins present at normal levels in PHF6 deficient cells?
6. Is the nucleolar localization of PHF6 important for the phenotype? Domain mapping related to the phenotype would be very interesting.
7. I find it interesting that a full 1/3 of the genes listed in Figure 1F are present in the SFARI database of genes mutated in autism patients. I realize this is not the topic of the paper but it is striking to me that there such a high enrichment and PHF6 is mutated in a related disorder (Börjeson-Forssman-Lehmann syndrome or BFLS).
8. Related to this point, BFLS patients that have mutant PHF6 do not exhibit pathologies typically associated with NHEJ defects (ex. SCID, Lig4 syndrome). This would suggest that PHF6 loss at the organismal level is not simply functionally equivalent to loss of NHEJ. This does not invalidate the results but could warrant discussion as the figure equates the defect with Ku loss and BFLS patients are not reported to be immunodeficient.
9. Further related to this point, this statement should be corrected: "...which is a rare X-linked genetic disorder characterized by mental retardation and craniopharyngeal abnormalities, and hematological cancers". BFLS is not associated with cancer, somatic, not hereditary mutations in PHF6 are to my understanding.

Minor points

1. I do not understand this sentence: "However, especially hits that led to increased recovery in U2OS behaved differently in RPE1 cells."
2. I will admit that I am being a bit semantic but the statement "However, the fact that 53BP1 was a hit in the screen nicely indicates that our screen allows for the identification of genes specially involved in DNA repair" is somewhat self serving as there is no evidence that this is why it comes as a hit in the screen (it is not included in the functional assays in Figure 2). While I agree that genes that influence NHEJ are identified, 53BP1 does have non-repair related roles in mitosis and potentially other functions and it is not clear that an NHEJ repair defect per se is what led to its identification.
3. BRD2 has been directly implicated in the transcriptional regulation of NHEJ genes:
PMID:29346775
4. SMARCA4, SMARCB1 and ACTL6A are all components of the SWI/SNF complex. This enrichment is not mentioned and their relationship not shown in Figure 1F.
5. While I appreciate that the authors are not engaging in gratuitous self-referencing, it is odd that the previous EMBO Reports paper from the Medema lab performing a G2 recovery screen using a kinome library is not mentioned, as it identified TLK2, a regulator of histone chaperones. This would seem to fit thematically with what the authors propose and ASF1a, that is also implicated in that study, has been shown to regulate NHEJ (PMID:28943310).

Referee #3:

This paper describes the discovery of PHF6, a chromatin regulator, as a component of NHEJ. They performed a RNAi screen in U2OS cells. Cells were synchronized in G2, irradiated and IF was used

to evaluate mitotic entry. They found 22 candidates among these chromatin modifiers as novel regulators of recovery from the DNA damage checkpoint arrest. They also found that PHF6 loss of function compromised the G2 checkpoint, PHF6 was recruited to sites of DNA damage in a PARP-dependent manner, and was needed for NHEJ.

Comments

- 1) The authors need to investigate the difference between A-NHEJ v C-NHEJ. PARP1 is also important for A-NHEJ. This can be done using this technique: EJ5-GFP can be used to distinguish these events since C-NHEJ, but not A-NHEJ, faithfully restores the I-SceI site. This probably applies to the CD4 reporter.
- 2) P53 is essential part of the phenotype for cells deleted for Ku or other NHEJ proteins as the mouse data with MEF shows and that could be the reason for the results presented in supplemental figure 3B.
- 3) Show a western blot to confirm the expression of GFP-PHF6 in fig. 3 D.

Cross-comments referee 1:

I think we all agree that some further analysis of PHF6 is needed, going from break accumulation to NHEJ pathway analysis. The authors are extremely competent, so this should not take more than 3 months. Complementation would in my opinion be important because the siRNAs and CRISPRs are far from uniform in phenotype. I am fine with authors selecting a few key PHF6 versions for complementation, and that they select a few assays for follow up (mitotic entry, 53BP1/γH2AX foci for example). The disease angle could be saved for the future as that would require time beyond 3 months.

Cross-comments referee 2:

To me the biggest concern amongst all reviewers is mechanism- particularly whether this is a transcriptional (indirect) effect on NHEJ or PHF6 plays a more direct role as inferred from the experiments presented by the authors. I would suggest that they validate expression of NHEJ factors in the knockdown, control for cell cycle and perform the EJ5 assays suggested by reviewer 3. Complementation would also be potentially informative but without knowing the targets of the different domains or whether it is a direct vs indirect effect, it may not provide clear insights into mechanism until the prior 2 experiments are done.

Cross-comments referee 3:

I agree with the comments of reviewer #2 and I believe this would be sufficient.

1st Revision - authors' response

27 September 2019

Point-by-point response to reviewer's comments

Referee #1:

Overview: Warmerdam et al. present a progressive genetic screening analysis to identify unappreciated factors required for recovery from the DNA damage G2 checkpoint. Not surprisingly, many screen hits were involved in DNA repair as their deficiency halts DNA damage checkpoint recovery due to deficient lesion clearance. The primary lead from this analysis was the implication of PHF6 in ensuring checkpoint recovery and DNA repair following DNA damage.

Novelty/Quality: The presented manuscript is well written by experts within the DDR/checkpoint field. Overall, it is of good quality with several complementary approaches (PHF6 loss-of-function

addressed with multiple siRNAs and CRISPR) and examines DDR/checkpoint signaling from a less appreciated viewpoint of recovery. Further, this manuscript indicates the disease relevant factor PHF6 as a DDR factor. However, though generally of solid quality, the current manuscript version is highly descriptive and offers little insight into the molecular function and mechanism of PHF6 in the DDR response.

We thank the reviewer for his/her positive and helpful comments and suggestions. The manuscript now contains additional experiments to further address the molecular mechanism of how PHF6 regulates the DDR and checkpoint recovery (see below).

Main points:

1) The authors primary supposition is that loss of PHF6 leads to a repair defect following DNA damage as indicated by the elevated levels of the marker phosphorylated-H2AX. To properly support this claim, they should include neutral COMET assay analysis to better reflect the extent of damage still present in the cells at this time.

A very good suggestion. We have performed neutral single cell electrophoresis (COMET assay) analysis (Figure 3H in the revised manuscript). The results indicate that cells lacking PHF6 have significantly more unrepaired DNA lesions in unperturbed conditions, and additionally show more DNA breaks after IR, suggesting that repair is less efficient in the absence of PHF6. These results are consistent with the data in Figure 3G showing that PHF6 knock out cells display increased numbers of gH2AX foci.

2) The authors should provide functional-mechanistic explanations for the role of PHF6. The genetic KO system is optimally suited to dissect the role of PHF6 through complementation, preferably also with disease variants. Such complementation should address major DDR aspects for major PHF6 functional mutants (PHF6 recruitment, checkpoint recovery and death, DNA damage accumulation by COMET assay, NHEJ and 53BP1 pathway function, DNA end resection performance).

To perform the suggested complementation studies in PHF6 knock out cells, we generated PHF6 mutant constructs, in which either the PHD1 or PHD2 domain was deleted (delta aa 17-131 and delta aa 212-329, respectively) (Expanded View Figure 4A). In addition, we generated a C99F mutant (296G>T), a BFLS causing mutation that is located in a conserved cysteine in PHD1 (Lower et al. 2002).

All PHF6 constructs were verified by Sanger-Sequencing and expressed in U2OS PHF6 KO cells (Rebuttal Figure 1), although with different efficiencies. Previously, Wang et al. reported that deletion of the PHD1 domain and introduction of C99F abrogated the localization of PHF6 to the nucleolus (Wang et al. 2013). However, we found that all our mutants localise both in the nucleus and nucleoli, similarly to wildtype PHF6 (Rebuttal Figure 1). We are not sure what causes the discrepancy between our results and those from Wang et al., although the cell type could play a role, as we have used U2OS cells, whereas HeLa cells were used in the other study. Due to these results, we decided to exclude the C99F mutant from our analysis, as we felt that at this moment, this mutant would not give us more information on the mechanism of how PHF6 controls checkpoint recovery.

Next, we determined gH2AX and 53BP1 IRIF in U2OS PHF6 KO cells expressing the PHD domain deletion mutants and wildtype GFP-PHF6. The wildtype PHF6 rescued the increase in IR-induced gH2AX foci that we observed in PHF6 KO cells and the PHD1 and PHD2 domain deletion mutants partially rescued this effect (Expanded View Figure 4C). Interestingly, complementing the PHF6 KO cells with PHF6 wildtype could rescue of the decreased number of 53BP1 foci in response to IR, whereas expression of the PHD domain mutants did not (Expanded View Figure 4D). Subsequent repair of DSBs (Expanded View Figure 4E) and checkpoint recovery (Expanded View Figure 4F – Due to logistic problems (defective X-ray source), the cells in this experiment were damaged by etoposide instead of IR) were also dependent on both PHD domains. Together these results indicate that both PHD domains are important for the role of PHF6 in NHEJ and recovery.

Previous reports indicated that depletion of PHF6 leads to increased levels of gH2AX (Wang et al. 2013, Van Vlierberghe et al. 2010). Wang et al. subscribe this to increased rRNA synthesis in cells depleted for PHF6 and subsequent formation of R-loops, resulting in DNA breaks (Wang et al.

2013). The PHD1 domain is associated with the function of PHF6 as a transcriptional repressor of rRNA synthesis. The PHD2 domain has so far only been reported to bind to double stranded DNA *in vitro* (Liu et al. 2014). Future research will elucidate the exact function of each of the PHD domains in PHF6.

Minor points:

Figure 4 legend indicates panel F, need to be change to E.

We have changed the legend of Figure 4 accordingly.

Authors need to indicate radiation dosages used for the screens. Starting with figure 1A.

The radiation dose for the screen is now indicated in the accompanying Figure legends.

Cross-comments referee 1:

I think we all agree that some further analysis of PHF6 is needed, going from break accumulation to NHEJ pathway analysis. The authors are extremely competent, so this should not take more than 3 months. Complementation would in my opinion be important because the siRNAs and CRISPRs are far from uniform in phenotype. I am fine with authors selecting a few key PHF6 versions for complementation, and that they select a few assays for follow up (mitotic entry, 53BP1/γH2AX foci for example). The disease angle could be saved for the future as that would require time beyond 3 months.

Please see answer above (point 2) on complementation. We have investigated checkpoint recovery and the DDR, by complementing PHF6 KO cells with PHF6 PHD1 and PHD2 deletion mutants, as suggested by the reviewer. The obtained results provide further insight into the mechanism of how PHF6 regulates DNA repair and recovery.

Expression of PHF6 mutants lacking either PHD domain in U2OS PHF6 KO cells indicated that both these domains are required for the DDR and checkpoint recovery. Both domains are highly conserved and the PHD1 domain has been reported to interact and negatively regulate UBF (Wang et al. 2013), an important factor in promoting ribosomal DNA transcription. PHD2 was reported to bind to double stranded DNA *in vitro* (Liu et al. 2014), thereby making it unique, as most PHD domains bind to histones. Interestingly, similar domains have been found in chromatin-associated proteins, some of which have also been implicated in the DDR (Liu et al. 2014). This suggests that PHF6 may modulate chromatin in response to DNA damage through its PHD domains, to repress transcription at break sites and thereby promote DNA repair. Future work will undoubtedly uncover in what way the PHD domains modulate the DDR and checkpoint recovery.

The importance of the PHD domains is highlighted by the mutations found in cancer, as the majority of mutations found in T-ALL are in the PHD domains (Liu et al. 2014), and about half of these are missense mutations (Van Vlierberghe et al. 2010), which could alter the functional characteristics of the protein, possibly leading to changes in DNA repair.

Albeit highly interesting, we agree with the reviewer that the disease angle is not within the scope of the current manuscript.

Rebuttal Figure 1:

A) Expression of GFP-PHF6 wildtype (WT), cytosine 99 to phenylalanine (C99F) and PHD1 and PHD2 domain deletion mutants (ΔPHD1 and ΔPHD2) in PHF6 KO cells. **B)** Localisation of PHF6 WT, C99F, ΔPHD1 and ΔPHD2 in PHF6 KO cells by direct fluorescence.

Referee #2:

The manuscript from Smits and colleagues presents results from a screen to identify chromatin proteins involved in recovery from the G2 checkpoint. Using a strategy the authors have exploited previously, they performed an siRNA screen on cells treated with IR and trapped in nocodazole, identifying numerous proteins that enhance or inhibit mitotic entry. These are validated in multiple cell lines to identify consistent hits and then sub analyzed as to whether they influence p53 activation or DNA damage signaling via ATM/ATR. They focus on several novel hits that include BRD2, ACTL6A and PHF6. They show that PHF6 depletion of knockout impairs mitotic entry after DNA damage, reduces 53BP1 foci and impairs NHEJ and that PHF6 localizes directly to DSBs.

Overall the manuscript is well presented and data of high quality. The results are interesting and novel, as to my knowledge PHF6 has not been implicated in DNA repair previously and the screening results may be of interest to many labs. My only criticism would be that the role of PHF6 in NHEJ remains unclear. While data presented suggests a potential direct role, its other functions are not ruled out (ex. transcription of NHEJ genes, as has been shown for BRD2) and no data demonstrates that the endogenous protein is actually recruited to DNA breaks.

We thank the reviewer for his/her positive comments and thoughtful suggestions. The manuscript now contains additional experiments to address whether PHF6 (in)directly affects NHEJ and thereby the DDR and checkpoint recovery.

Main points:

- 1) The manuscript culminates in a focus on PHF6 from Figure 3 on. In the PHF6 knockdown or knockout cells, is there a defect in cell cycle progression in untreated cells? Does this influence the appearance of 53BP1 foci that are cell cycle dependent?*
- 2) Cell cycle status also influences NHEJ, it would be important to show data regarding the effects of PHF6 loss in order to properly evaluate the data in figure 3 and 4.*

We addressed this point by analyzing the cell cycle profiles after depletion of PHF6 by siRNA and in PHF6 KO and control cells and have included the data as Expanded View Figure 3F and G. Loss or downregulation of PHF6 did not result into significant changes in cell cycle distribution. We therefore conclude that the effect of PHF6 on IR-induced focus formation of 53BP1 is independent of the cell cycle.

- 3) Why is co-localization with NBS1 investigated vs factors involved in NHEJ?*

NBS1 is a component of the MRE11-Rad50-NBS1 complex, that is also involved in NHEJ (Zha, Boboila & Alt 2009). Besides, NBS1 is an early DNA damage response marker which recruitment kinetics to laser-stripes have been studied extensively (Lukas et al. 2003, Luijsterburg et al. 2009, Luijsterburg et al. 2016). Therefore NBS1 is often used as a positive control in laser-stripe experiments in the lab and this is also the reason why we included NBS1 in this analysis.

- 4) Does treating PHF6 KOs with DNA-PKcs inhibitor influence sensitivity or is it epistatic?*

This is an interesting experiment. Instead of using the suggested DNA-PKcs inhibitor, we addressed this point by depleting XRCC4, a NHEJ core component, by siRNA, in control and PHF6 KO cells and subsequent analysis of clonogenic survival upon IR. The results shown in Figure 4D and Expanded View Figure 5E indicate that loss of PHF6 and XRCC4 result in a similar sensitivity to IR. In addition, depleting XRCC4 in PHK6 KO cells did not lead to increased sensitivity to IR. These data confirm that these factors protect cells against DSBs by acting in the same pathway.

- 5) Are NHEJ proteins present at normal levels in PHF6 deficient cells?*

We have examined the protein levels of a broad range of NHEJ-associated proteins in U2OS cells depleted for PHF6 by siRNA and PHF6 KO and the (parental) controls cells by western blotting, as shown in Expanded View Figure 5D. The results indicate that the protein levels of Mre11, Nbs1, Rad50, Ku86 (Expanded View Figure 3C), Ku70, DNA-PKcs, XRCC4, Ligase IV, Artemis and XLF did not change after depletion or loss of PHF6. Moreover, the protein levels of 53BP1, CtIP,

TopBP1, Rif1 and PARP1 were also unaffected after depletion and knockout of PHF6. Together these results strongly suggest that PHF6 does not control NHEJ by transcriptionally regulating the factors involved.

6) Is the nucleolar localization of PHF6 important for the phenotype? Domain mapping related to the phenotype would be very interesting.

We attempted to address this point by expressing a mutant of PHF6 containing a G>T mutation in nucleotide 294 that results in an amino acid change in cytosine 99 to phenylalanine (C>F) in U2OS PHF6 KO cells. The C99F amino acid substitution leads to BFLS (Lower et al. 2002, Cheng et al. 2018), is possibly involved in the interaction with upstream binding factor (UBF), and was reported not to localize in the nucleolus (Wang et al. 2013). In addition, we generated a PHF6 mutant in which the PHD1 domain was mutated, as this domain was reported to be important for the nucleolar localization (Wang et al. 2013). However, we did not observe differences in nucleolar localization of these mutants as compared to wildtype PHF6 (also see response to point 2 and cross-comments of reviewer #1 and Rebuttal Figure 1). We were therefore unfortunately not able to address whether the nucleolar function of PHF6 is important for checkpoint recovery.

7) I find it interesting that a full 1/3 of the genes listed in Figure 1F are present in the SFARI database of genes mutated in autism patients. I realize this is not the topic of the paper but it is striking to me that there such a high enrichment and PHF6 is mutated in a related disorder (Börjeson-Forssman-Lehmann syndrome or BFLS).

Indeed, a very interesting observation. Maybe future research could demonstrate a link between these disorders.

8) Related to this point, BFLS patients that have mutant PHF6 do not exhibit pathologies typically associated with NHEJ defects (ex. SCID, Lig4 syndrome). This would suggest that PHF6 loss at the organismal level is not simply functionally equivalent to loss of NHEJ. This does not invalidate the results but could warrant discussion as the figure equates the defect with Ku loss and BFLS patients are not reported to be immunodeficient.

The mutations in PHF6 found in BFLS might not affect the repair function of PHF6, and therefore, BFLS patients might not show immunological defects. However, PHF6 (conditional) knockout mice show a (mild) expansion of hematopoietic stem cells (Cheng et al. 2018, McRae et al. 2019, Miyagi et al. 2019). This could be (partly) explained by the transcriptional repressor function of PHF6, but decreased NHEJ and inefficient DDR/checkpoint recovery could possibly also play a role here. The study of BFLS and cancer-associated PHF6 mutations in DNA repair are obviously an interesting topic for future research.

9) Further related to this point, this statement should be corrected: "...which is a rare X-linked genetic disorder characterized by mental retardation and craniopharyngeal abnormalities, and hematological cancers". BFLS is not associated with cancer, somatic, not hereditary mutations in PHF6 are to my understanding.

According to Chao et al., BFLS may also represent as a cancer predisposition syndrome (Chao et al. 2010). We have added the reference to support the statement.

Minor points will be addressed through textual changes.

1) I do not understand this sentence: "However, especially hits that led to increased recovery in U2OS behaved differently in RPE1 cells."

To make this point more clear, the sentence has been changed to: "However, many of the hits that led to increased recovery in U2OS actually showed an opposite response in RPE-1 cells".

2) I will admit that I am being a bit semantic but the statement "However, the fact that 53BP1 was a hit in the screen nicely indicates that our screen allows for the identification of genes specially involved in DNA repair" is somewhat self-serving as there is no evidence that this is why it comes as a hit in the screen (it is not included in the functional assays in Figure 2). While I agree that genes

that influence NHEJ are identified, 53BP1 does have non-repair related roles in mitosis and potentially other functions and it is not clear that an NHEJ repair defect per se is what led to its identification.

The referee is right and we therefore re-phrased the sentence.

3) BRD2 has been directly implicated in the transcriptional regulation of NHEJ genes: PMID:29346775

We agree with the reviewer that bromodomain-containing (BRD) proteins have been implicated in the regulation of gene expression. BRD2 has been shown to be involved in regulation of transcribed genes (LeRoy, Rickards & Flint 2008). The aforementioned publication (Li et al. 2018) indicates the functional relevance of this regulation in prostate cancer, indicated by the finding that BRD4 (and BRD2) loss or inhibition influence NHEJ-mediated repair of DSBs. On the other hand, Floyd et al. showed that BRD4 can also act as a chromatin insulator and thereby influences the DDR more directly (Floyd et al. 2013). Moreover, BRD2 was shown to be recruited to chromatin surrounding DSBs, leading to the recruitment of 53BP1 and thereby promoting repair through NHEJ (Gursoy-Yuzugullu, Carman & Price 2017). These data indicate that BRD proteins are regulators of chromatin and thereby can influence chromatin-associated processes including transcription and DNA repair. Although highly interesting, the manuscript mostly focusses on the role of PHF6 and its involvement in NHEJ, a discussion about the role of BRD2 therefore seems unfitting and better suited for future work more directed towards this issue.

4) SMARCA4, SMARCB1 and ACLT6A are all components of the SWI/SNF complex. This enrichment is not mentioned and their relationship not shown in Figure 1F.

The reviewer is right that these proteins are components of the SWI/SNF complex. However, based on the results from Figure 1D and E, depletion of SMARCA4 did not meet the criteria to be identified as a hit. Therefore, SMARCA4 was not included in the analysis in Figure 1F, in which we only depicted complexes that contain multiple hits.

5) While I appreciate that the authors are not engaging in gratuitous self-referencing, it is odd that the previous EMBO Reports paper from the Medema lab performing a G2 recovery screen using a kinome library is not mentioned, as it identified TLK2, a regulator of histone chaperones. This would seem to fit thematically with what the authors propose and ASF1a, that is also implicated in that study, has been shown to regulate NHEJ (PMID:28943310).

We agree that the identification of TLK2 and ASF1a regulating checkpoint recovery (Bruinsma et al. 2016), and the data describing ASF1a to promote NHEJ (Lee et al. 2017) nicely fit our data. We have now discussed these papers in the revised manuscript.

Cross-comments referee 2:

To me the biggest concern amongst all reviewers is mechanism- particularly whether this is a transcriptional (indirect) effect on NHEJ or PHF6 plays a more direct role as inferred from the experiments presented by the authors. I would suggest that they validate expression of NHEJ factors in the knockdown, control for cell cycle and perform the EJ5 assays suggested by reviewer 3. Complementation would also be potentially informative but without knowing the targets of the different domains or whether it is a direct vs indirect effect, it may not provide clear insights into mechanism until the prior 2 experiments are done.

To address this point we have:

- i) Thoroughly investigated the protein levels of NHEJ proteins by western blot after depletion and loss of PHF6 (see referee #2, point 4)
- ii) Examined the cell cycle distribution after depletion and knockout of PHF6 (see referee #2, point 1)
- iii) Determined classical vs. alternative-NHEJ in GC92 cells (see referee #3, point 1)
- iv) Studied the contribution of the PHD domains of PHF6 in the DDR and recovery (see referee #1, point 2 and cross-comments)

Referee #3:

This paper describes the discovery of PHF6, a chromatin regulator, as a component of NHEJ. They performed a RNAi screen in U2OS cells. Cells were synchronized in G2, irradiated and IF was used to evaluate mitotic entry. They found 22 candidates among these chromatin modifiers as novel regulators of recovery from the DNA damage checkpoint arrest. They also found that PHF6 loss of function compromised the G2 checkpoint, PHF6 was recruited to sites of DNA damage in a PARP-dependent manner, and was needed for NHEJ.

We thank the reviewer for his/her positive and helpful comments and suggestions.

Main points:

1) The authors need to investigate the difference between A-NHEJ v C-NHEJ. PARP1 is also important for A-NHEJ. This can be done using this technique: EJ5-GFP can be used to distinguish these events since C-NHEJ, but not A-NHEJ, faithfully restores the I-SceI site. This probably applies to the CD4 reporter.

This was a good suggestion. To address this, we have measured classical vs. alternative-NHEJ by junction analysis in GC92 cells after I-SceI break-induction and subsequent repair. PHF6 was depleted using two different siRNA's, after which I-SceI was induced. Subsequently, genomic DNA was extracted and the region around the I-SceI cut-site was amplified through PCR, cloned into the pGEM T-easy vector and colonies were analyzed by Sanger-Sequencing (Taty-Taty et al. 2016, Schimmel et al. 2017). Repair was quantified by determining the extent of perfectly re-ligated breaks, and the ones containing insertions, deletions or both (delins) around the break-site (Figure 4E and Expanded View Figure 5F and G). The results show that depletion of PHF6 resembles junctional signatures observed after loss of Ku80, a critical factor in classical NHEJ. In control cells around 80% of the junctions are perfectly re-ligated and only 20% have mutations. This distribution shifts after depletion of Ku80 (~60% perfect re-ligation vs ~40% mutated junctions) or PHF6 (~70% perfect re-ligation vs ~30% mutated junctions). The mutated junctions after depletion of Ku80 or PHF6 show larger deletions and use larger microhomology. Taken together, these results indicate that classical NHEJ is impaired in cells depleted for PHF6, shifting repair to alternative NHEJ, which is characterized by larger deletions and increased microhomology.

2) p53 is an essential part of the phenotype for cells deleted for Ku or other NHEJ proteins as the mouse data with MEFs shows and that could be the reason for the results presented in supplemental figure 3B.

We thank the reviewer for this suggestion. Loss of p53 often synergizes with the loss of NHEJ as shown by Difilippantonio et al (Difilippantonio et al. 2000), thereby providing a possible explanation for the differential responses observed with the four PHF6 siRNA oligonucleotides in Figure 3B. We have added the following sentence to the manuscript: "Potential synergistic loss of p53 with knockdown of PHF6 using a particular siRNA oligonucleotide, as reported before for genes involved in DNA repair (Difilippantonio et al. 2000), could also result in a differential response".

3) Show a western blot to confirm the expression of GFP-PHF6 in fig. 3 D.

We now provide an accompanying western blot in Figure 3E, showing the expression of GFP-PHF6 in PHF6 KO cells. Likewise, we included western blots of expression of the PHD1 and PHD2 deletion mutants in PHF6 KO cells in Expanded View Figure 4B.

Cross-comments referee 3:

I agree with the comments of reviewer #2 and I believe this would be sufficient.

Please see response to cross-comments referee 2.

References

Chao, M.M., Todd, M.A., Kontny, U., Neas, K., Sullivan, M.J., Hunter, A.G., Picketts, D.J. & Kratz, C.P. 2010, "T-cell acute lymphoblastic leukemia in association with Borjeson-Forssman-

- Lehmann syndrome due to a mutation in PHF6", *Pediatric blood & cancer*, vol. 55, no. 4, pp. 722-724.
- Cheng, C., Deng, P.Y., Ikeuchi, Y., Yuede, C., Li, D., Rensing, N., Huang, J., Baldrige, D., Maloney, S.E., Dougherty, J.D., Constantino, J., Jahani-Asl, A., Wong, M., Wozniak, D.F., Wang, T., Klyachko, V.A. & Bonni, A. 2018, "Characterization of a Mouse Model of Borjeson-Forssman-Lehmann Syndrome", *Cell reports*, vol. 25, no. 6, pp. 1404-1414.e6.
- Difilippantonio, M.J., Zhu, J., Chen, H.T., Meffre, E., Nussenzweig, M.C., Max, E.E., Ried, T. & Nussenzweig, A. 2000, "DNA repair protein Ku80 suppresses chromosomal aberrations and malignant transformation", *Nature*, vol. 404, no. 6777, pp. 510-514.
- Floyd, S.R., Pacold, M.E., Huang, Q., Clarke, S.M., Lam, F.C., Cannell, I.G., Bryson, B.D., Rameseder, J., Lee, M.J., Blake, E.J., Fydrych, A., Ho, R., Greenberger, B.A., Chen, G.C., Maffa, A., Del Rosario, A.M., Root, D.E., Carpenter, A.E., Hahn, W.C., Sabatini, D.M., Chen, C.C., White, F.M., Bradner, J.E. & Yaffe, M.B. 2013, "The bromodomain protein Brd4 insulates chromatin from DNA damage signalling", *Nature*, vol. 498, no. 7453, pp. 246-250.
- Gursoy-Yuzugullu, O., Carman, C. & Price, B.D. 2017, "Spatially restricted loading of BRD2 at DNA double-strand breaks protects H4 acetylation domains and promotes DNA repair", *Scientific reports*, vol. 7, no. 1, pp. 12921-017-13036-5.
- Lee, K.Y., Im, J.S., Shibata, E. & Dutta, A. 2017, "ASF1a Promotes Non-homologous End Joining Repair by Facilitating Phosphorylation of MDC1 by ATM at Double-Strand Breaks", *Molecular cell*, vol. 68, no. 1, pp. 61-75.e5.
- LeRoy, G., Rickards, B. & Flint, S.J. 2008, "The double bromodomain proteins Brd2 and Brd3 couple histone acetylation to transcription", *Molecular cell*, vol. 30, no. 1, pp. 51-60.
- Li, X., Baek, G., Ramanand, S.G., Sharp, A., Gao, Y., Yuan, W., Welti, J., Rodrigues, D.N., Dolling, D., Figueiredo, I., Sumanasuriya, S., Crespo, M., Aslam, A., Li, R., Yin, Y., Mukherjee, B., Kanchwala, M., Hughes, A.M., Halsey, W.S., Chiang, C.M., Xing, C., Raj, G.V., Burma, S., de Bono, J. & Mani, R.S. 2018, "BRD4 Promotes DNA Repair and Mediates the Formation of TMPRSS2-ERG Gene Rearrangements in Prostate Cancer", *Cell reports*, vol. 22, no. 3, pp. 796-808.
- Liu, Z., Li, F., Ruan, K., Zhang, J., Mei, Y., Wu, J. & Shi, Y. 2014, "Structural and functional insights into the human Borjeson-Forssman-Lehmann syndrome-associated protein PHF6", *The Journal of biological chemistry*, vol. 289, no. 14, pp. 10069-10083.
- Lower, K.M., Turner, G., Kerr, B.A., Mathews, K.D., Shaw, M.A., Gedeon, A.K., Schelley, S., Hoyme, H.E., White, S.M., Delatycki, M.B., Lampe, A.K., Clayton-Smith, J., Stewart, H., van Ravenswaay, C.M., de Vries, B.B., Cox, B., Grompe, M., Ross, S., Thomas, P., Mulley, J.C. & Gecz, J. 2002, "Mutations in PHF6 are associated with Borjeson-Forssman-Lehmann syndrome", *Nature genetics*, vol. 32, no. 4, pp. 661-665.
- Luijsterburg, M.S., de Krijger, I., Wiegant, W.W., Shah, R.G., Smeenk, G., de Groot, A.J.L., Pines, A., Vertegaal, A.C.O., Jacobs, J.J.L., Shah, G.M. & van Attikum, H. 2016, "PARP1 Links CHD2-Mediated Chromatin Expansion and H3.3 Deposition to DNA Repair by Non-homologous End-Joining", *Molecular cell*, vol. 61, no. 4, pp. 547-562.
- Luijsterburg, M.S., Dinant, C., Lans, H., Stap, J., Wiernasz, E., Lagerwerf, S., Warmerdam, D.O., Lindh, M., Brink, M.C., Dobrucki, J.W., Aten, J.A., Foustari, M.I., Jansen, G., Dantuma, N.P., Vermeulen, W., Mullenders, L.H., Houtsmuller, A.B., Verschure, P.J. & Driel, R.v. 2009, "Heterochromatin protein 1 is recruited to various types of DNA damage", *The Journal of cell biology*, vol. 185, no. 4, pp. 577-586.
- Lukas, C., Falck, J., Bartkova, J., Bartek, J. & Lukas, J. 2003, "Distinct spatiotemporal dynamics of mammalian checkpoint regulators induced by DNA damage", *Nature cell biology*, vol. 5, no. 3, pp. 255-260.
- McRae, H.M., Garnham, A.L., Hu, Y., Witkowski, M.T., Corbett, M.A., Dixon, M.P., May, R.E., Sheikh, B.N., Chiang, W., Kueh, A.J., Nguyen, T.A., Man, K., Gloury, R., Aubrey, B.J., Policheni, A., Di Rago, L., Alexander, W.S., Gray, D.H.D., Strasser, A., Hawkins, E.D., Wilcox, S., Gecz, J., Kallias, A., McCormack, M.P., Smyth, G.K., Voss, A.K. & Thomas, T. 2019, "PHF6 regulates hematopoietic stem and progenitor cells and its loss synergizes with expression of TLX3 to cause leukemia", *Blood*, vol. 133, no. 16, pp. 1729-1741.
- Miyagi, S., Sroczynska, P., Kato, Y., Nakajima-Takagi, Y., Oshima, M., Rizq, O., Takayama, N., Saraya, A., Mizuno, S., Sugiyama, F., Takahashi, S., Matsuzaki, Y., Christensen, J., Helin, K. & Iwama, A. 2019, "The chromatin-binding protein Phf6 restricts the self-renewal of hematopoietic stem cells", *Blood*, vol. 133, no. 23, pp. 2495-2506.

- Schimmel, J., Kool, H., van Schendel, R. & Tijsterman, M. 2017, "Mutational signatures of non-homologous and polymerase theta-mediated end-joining in embryonic stem cells", *The EMBO journal*, vol. 36, no. 24, pp. 3634-3649.
- Taty-Taty, G.C., Chailleux, C., Quaranta, M., So, A., Guirouilh-Barbat, J., Lopez, B.S., Bertrand, P., Trouche, D. & Canitrot, Y. 2016, "Control of alternative end joining by the chromatin remodeler p400 ATPase", *Nucleic acids research*, vol. 44, no. 4, pp. 1657-1668.
- Van Vlierberghe, P., Palomero, T., Khiabani, H., Van der Meulen, J., Castillo, M., Van Roy, N., De Moerloose, B., Philippe, J., Gonzalez-Garcia, S., Toribio, M.L., Taghon, T., Zuurbier, L., Cauwelier, B., Harrison, C.J., Schwab, C., Pisecker, M., Strehl, S., Langerak, A.W., Gecz, J., Sonneveld, E., Pieters, R., Paietta, E., Rowe, J.M., Wiernik, P.H., Benoit, Y., Soulier, J., Poppe, B., Yao, X., Cordon-Cardo, C., Meijerink, J., Rabadan, R., Speleman, F. & Ferrando, A. 2010, "PHF6 mutations in T-cell acute lymphoblastic leukemia", *Nature genetics*, vol. 42, no. 4, pp. 338-342.
- Wang, J., Leung, J.W., Gong, Z., Feng, L., Shi, X. & Chen, J. 2013, "PHF6 regulates cell cycle progression by suppressing ribosomal RNA synthesis", *The Journal of biological chemistry*, vol. 288, no. 5, pp. 3174-3183.
- Zha, S., Boboila, C. & Alt, F.W. 2009, "Mre11: roles in DNA repair beyond homologous recombination", *Nature structural & molecular biology*, vol. 16, no. 8, pp. 798-800.

2nd Editorial Decision

28 October 2019

Thank you for the submission of your revised manuscript. We have now received the comments from all referees, and I am happy to say that all support its publication now. We can therefore in principle accept your manuscript.

Only a few minor changes are still required:

- Please address all remaining referee concerns in the final manuscript file.
- Please send us a correct conflict of interest statement.
- The abstract, or better the novel findings, need to be described in present tense, please correct.
- I attach to this email a manuscript word file with tracked changes by our data editors. Please address all comments using the track changes option and send us back a corrected file with your final manuscript submission.
- The EV figure legends need to be included at the end of the main manuscript file.
- As far as I can see, the Appendix only contains a table. This table can be called Table EV2 instead. It might be easier if you submit both EV tables as excel files. If you want to keep the Appendix, it needs a title and table of content with page numbers.
- Given that you have calculated statistics in several of your figures, the first part of the author checklist needs to be completed. Please send us a new checklist with the missing information filled in.
- It would be better if the scale bars in the microscopy images could be a little bigger. Ideally, they should be readable at 100% image size.
- Please try to avoid over-contrasting the blots, eg in figures 3, EV3, EV4 some blots are over-contrasted.

Given that you as corresponding author are in the Netherlands, you are probably eligible for publication of your article in the open access format in a way that is free of charge for the authors. Please contact either the administration at your institution or our publishers at Wiley (emboreports@wiley.com) for further questions. The general information of Wiley's agreements with the Netherlands can be found here: <https://authorservices.wiley.com/author-resources/Journal-Authors/open-access/affiliation-policies->

payments/vsnu-agreement.html

EMBO press papers are accompanied online by A) a short (1-2 sentences) summary of the findings and their significance, B) 2-3 bullet points highlighting key results and C) a synopsis image that is 550x200-400 pixels large (the height is variable). You can either show a model or key data in the synopsis image. Please note that text needs to be readable at the final size. Please send us this information along with the submission of your revised manuscript.

REFEREE REPORTS

Referee #1:

The authors have fully addressed all the points raised in my review.

Referee #2:

The authors have fully addressed the priorities established by the reviewers with new experiments and responded to all of the comments to clarify other issues. I am now satisfied that the primary conclusions of the manuscript are fully supported. I wanted to point out only 2 minor details that I noticed upon reading:

1. In Figure 1F they label the NuA4 complex and then label the other Cohesion. The name of the complex is Cohesin, and it of course plays a role in chromosome cohesion, but to be consistent with the other that uses the complex name, consider correcting this.
2. In Exp Figure 5D in the PHF6 KO cells, there is a clear band of the correct size. Why is this? Should these cells not be considered KO but rather mutant? Is it clear what this is?

Referee #3:

the authors have answered all my comments and i believe the paper is ready for publication

2nd Revision - authors' response

4 November 2019

Remaining referee concerns

Referee #2: *The authors have fully addressed the priorities established by the reviewers with new experiments and responded to all of the comments to clarify other issues. I am now satisfied that the primary conclusions of the manuscript are fully supported. I wanted to point out only 2 minor details that I noticed upon reading:*

1. *In Figure 1F they label the NuA4 complex and then label the other Cohesion. The name of the complex is Cohesin, and it of course plays a role in chromosome cohesion, but to be consistent with the other that uses the complex name, consider correcting this.*

We thank the reviewer for noticing and have relabeled Cohesion in Figure 1F to Cohesin.

2. *In Exp Figure 5D in the PHF6 KO cells, there is a clear band of the correct size. Why is this? Should these cells not be considered KO but rather mutant? Is it clear what this is?*

We would like to point out that Figure EV3C shows an aspecific band, which is running just under the PHF6 protein. Depending on the percentage of gel, this band is separated from the PHF6 protein (as in EV3C), or running at a similar height (such as in EV5D). Other figures, for example EV5E, clearly show that the PHF6 protein is absent in the KO cells. We have included a short explanation in the legends of Figures 3E and EV5D.

Corresponding Author Name: Daniël O. Warmerdam

Manuscript Number: EMBOR-2019-48460V1